# Consequences of permafrost degradation for Arctic infrastructure - bridging the model gap between regional and engineering scales

Thomas Schneider von Deimling[1,2], Hanna Lee[3], Thomas Ingeman-Nielsen[4], Sebastian Westermann[5], Vladimir Romanovsky[6], Scott Lamoureux[7], Donald A. Walker[8], Sarah Chadburn[9], Erin Trochim[10], Lei Cai[11], Jan Nitzbon[1,2], Stephan Jacobi[1], Moritz Langer[1,2]

[1]Alfred Wegener Institute Helmholtz Centre for Polar and Marine Research, 14473 Potsdam Germany
[2]Humboldt University of Berlin, Geography Department, Unter den Linden 6, 10099 Berlin, Germany
[3]NORCE Norwegian Research Centre, Bjerknes Centre for Climate Research, Bergen, Norway
[4]Department of Civil Engineering, Technical University of Denmark, 2800 Kgs. Lyngby, Denmark
[5]University of Oslo, Department of Geosciences, Sem Sælands vei 1, 0316 Oslo, Norway
[6]University of Alaska Fairbanks, Geophysical Institute, Fairbanks, Alaska, USA
[7]Queen's University, Department of Geography and Planning, Kingston, ON  K7L 3N6, Canada
[8]University of Alaska Fairbanks, Institute of Arctic Biology, Department of Biology and Wildlife, Fairbanks, Alaska, USA
[9]University of Exeter, College of Engineering, Mathematics and Physical Sciences, Exeter EX4 4QE, UK
[10]University of Alaska Fairbanks, Alaska Center for Energy and Power, Fairbanks, Alaska, USA
[11]Department of Atmospheric Sciences, Yunnan University, Kunming 650034, China

*Correspondence to*: Thomas Schneider von Deimling (thomas.schneider@awi.de)

**Abstract.** Infrastructure built on perennially frozen ice-rich ground relies heavily on thermally stable subsurface conditions. Climate warming-induced deepening of ground thaw puts such infrastructure at risk of failure. For better assessing the risk of large-scale future damage to Arctic infrastructure, improved strategies for model-based approaches are urgently needed.

We used the laterally-coupled one-dimensional heat conduction model CryoGrid3 to simulate permafrost degradation affected by linear infrastructure. We present a case study of a gravel road built on continuous permafrost (Dalton highway, Alaska) and forced our model under historical and strong future warming conditions (following the RCP8.5 scenario). As expected, the presence of a gravel road in the model leads to higher net heat flux entering the ground compared to a reference run without infrastructure, and thus a higher rate of thaw.  Further, our results suggest that road failure is likely a consequence of lateral destabilization due to talik formation in the ground beside the road, rather than a direct consequence of a top-down thawing and deepening of the active layer below the road centre. In line with previous studies, we identify enhanced snow accumulation and ponding (both a consequence of infrastructure presence) as key factors for increased soil temperatures and road degradation. Using differing horizontal model resolutions we show that it is possible to capture these key factors and their

impact on thawing dynamics with a low number of lateral model units, underlining the potential of our model approach for use in pan-arctic risk assessments.

Our results suggest a general two-phase behaviour of permafrost degradation: an initial phase of slow and gradual thaw, followed by a strong increase in thawing rates after exceedance of a critical ground warming. The timing of this transition and the magnitude of thaw rate acceleration differ strongly between undisturbed tundra and infrastructure-affected permafrost ground. Our model results suggest that current model-based approaches which do not explicitly take into account infrastructure in their designs are likely to strongly underestimate the timing of future Arctic infrastructure failure.

By using a laterally-coupled one-dimensional model to simulate linear infrastructure, we infer results in line with outcomes from more complex 2D- and 3D-models, but our model's computational efficiency allows us to account for long-term climate change impacts on infrastructure from permafrost degradation. Our model simulations underline that it is crucial to consider climate warming when planning and constructing infrastructure on permafrost as a transition from a stable to a highly unstable state can well occur within the service life time (about 30 years) of such a construction. Such a transition can even be triggered in the coming decade by climate change for infrastructure built on high northern latitude continuous permafrost that displays cold and relatively stable conditions today.

## 1    Introduction

Land surface temperatures in the Arctic are reported to have warmed by more than 0.5 °C per decade since 1981 (Comiso and Hall, 2014). This exceeds the average global warming by a factor between two to three and already is leading to pronounced observable cryospheric changes ranging from sea-ice decline, Greenland ice sheet melt and reduction in spring snow cover, to a wide range of permafrost degradation processes (Rowland et al., 2010). Rapid changes of permafrost landscapes are in particular triggered and accelerated by melting of excess ground ice, resulting in land surface subsidence (Kokelj and Jorgenson, 2013). The potential damage to ecosystems and infrastructure caused by permafrost thaw is linked to diverse ecological, social, and financial risks. The economic development of the Arctic requires highly resilient infrastructure such as supply roads, pipelines, fuel storages, airports, and other buildings to be constructed on highly sensitive frozen ground (Larsen and Fondahl, 2015). Even offshore activities and shipping in the Arctic depend on reliable onshore facilities in permafrost-dominated regions such as ports and local transportation networks. Functional and safe infrastructure is important for the livelihood of the local population, and is directly dependent on the thermal stability of the underlying and surrounding permafrost (Romanovsky, 2017).

Field investigations have demonstrated that the degradation of permafrost landscapes is not limited to top-down soil warming and thawing, but is often accompanied by thermokarst and a variety of erosional and mass wasting processes that operate on different spatial and temporal scales (Fortier et al., 2007;Liljedahl et al., 2016). Landslides, thermokarst features, and thermo-

erosion gullies are prominent geomorphological features that reflect the active dynamics of destabilized permafrost landscapes, which directly threaten man made infrastructure (Nelson et al., 2001). Infrastructure itself can also cause thermokarst and other processes resulting from thawing ground ice. Significant efforts are made during road planning to quantify ground ice (Trochim et al., 2016). Ground ice distribution is one important factor towards understanding and predicting thermokarst processes that affect infrastructure. Additionally, these thermokarst processes in turn affect local heat, water, erosion and carbon and nutrient exchange processes, so that a variety of feedback mechanisms such as additional greenhouse gas emissions may well be associated with permafrost erosion (Chapin et al., 2005;Schuur et al., 2015;Schneider von Deimling et al., 2015;Walter Anthony et al., 2018). Climate, terrain and local processes are critical additions to generate holistic understanding of infrastructure response to permafrost (Stephani et al., 2014). Reliable and prompt assessments of risks of possible damages to both the ecology and infrastructure are therefore critically important.

Recent studies indicate that many areas are already experiencing permafrost-related issues affecting engineered structures, including Alaska where this study is focused. 18% (34 out of 187) of rural communities evaluated there were designated as high risk with over half located in continuous permafrost (Kanevskiy et al., 2019). In the oilfield region of Prudhoe Bay on the North Slope of Alaska, permafrost degradation has been documented over 62 years with acceleration after 1990 in the rates of thermokarst related to a rise in summer air and permafrost temperatures (Raynolds et al., 2014). Considering projected climate change under the RCP8.5 scenario, Streletskiy et al. (2019) and Suter et al. (2019) show that infrastructure built on permafrost will be negatively affected by decreases in bearing capacity and ground subsidence by mid-century, resulting in large infrastructure lifecycle replacement costs. The risk of disasters caused by damage to sensitive infrastructure such as pipelines, fuel storages, and plants have yet to be estimated. Passive cooling techniques can be effective at reducing climate effects but mitigation and adaptation can be more expensive than conventional construction (Doré et al., 2016). Globally, the emission of additional greenhouse gases from thawing permafrost is increasing the risk for extra costs of overall climate change (Hope and Schaefer, 2016;Melvin et al., 2017).

Given the lack of high-resolution circum-Arctic model projections of permafrost thaw and subsidence, recent risk assessment studies of infrastructure failure have used statistical approaches and suggested that fundamental engineering structures in the Arctic are at risk by mid-century (Hjort et al., 2018;Karjalainen et al., 2019). Such statistical approaches allow accounting for various sources of local-scale information such as ground ice content, sediment type, or slope gradient at comparatively low computational costs. They have their strength in providing high resolution geospatial maps for evaluating the potential of future infrastructure failure in specific regions. But these statistical approaches cannot capture transient changes in permafrost well enough and they particularly fail to predict dynamical and non-linear features of permafrost degradation (such as thermokarst). Therefore these approaches can miss key factors that determine the timing of future infrastructure damage, which likely lead to an underestimation of the timing when infrastructure is at risk in the future. Daanen et al. (2011) have suggested a risk assessment approach that can benefit frozen ground engineering applications. Their approach accounts for dynamical permafrost modelling, driven by regional climate change predictions, to estimate a Permafrost Thaw Potential which includes

the effects of climate warming and infrastructure elements on future thawing depths but does not account for lateral interactions between infrastructure and adjacent permafrost ground. Different approaches used in previous studies have their strengths and weaknesses, outlined below.

## 1.1 State-of-the-art global modelling of permafrost degradation

The current global land surface models (LSMs) have undergone extensive improvements within the last decade to include key processes related to permafrost changes and degradation. The focus of these model developments is on a better description of physical thaw processes and of biogeochemical cycles of soil carbon release, but the impact of infrastructure has not yet been included.

Widespread thawing of permafrost is expected in a warmer future climate and LSM-based model studies suggest large-scale degradation of near-surface permafrost at the end of the 21st century (Lawrence et al., 2008;Lawrence et al., 2011; McGuire et al, 2018). However, current modelling approaches used to simulate permafrost degradation under a warming climate are highly simplistic since they only consider one-dimensional (top-down) thawing and ignore important lateral processes such as snow redistribution, soil erosion and mass wasting, subsurface water flow - common processes in many regions that accelerate thaw. This limits the ability to understand the magnitude of impacts under permafrost thaw. Thus, current model assessments are most likely     underestimating permafrost thaw impacts, which is underlined by observational evidence of strong permafrost degradation even under present day climate conditions not being captured by models (Farquharson et al., 2019;Nitzbon et al., 2020).

## 1.2 Modelling climate change impacts on infrastructure built on permafrost - an issue of scales

Under the current projections of climate warming in the Arctic, there is an increasing need to incorporate these permafrost thaw scenarios into understanding the fate of infrastructures in the Arctic region. Given the pace of permafrost degradation, Arctic infrastructure may well be seriously affected within 30 years, a common service lifespan for many infrastructure types. It is important to note that infrastructure constructed on permafrost is not only being impacted by climate change, but it is itself affecting ground thermal and hydrological conditions, thus leading to infrastructure-related impacts on permafrost stability. Infrastructure embankments strongly change ground thermal properties and surface albedo. They result in an increased winter insulation where additional snow accumulates (e.g. at the embankment shoulder and toe, (Fortier et al., 2011;O'Neill and Burn, 2017). In contrast, snow clearance at the road surface leads to strong subsurface winter cooling. Infrastructure embankments can also act as a dam that alters natural drainage networks, causing water to accumulate next to the road (Andersland and Ladanyi, 1994;de Grandpre et al., 2012;O'Neill and Burn, 2017). An elevated permafrost table in the embankment also hampers drainage, which may result in ponding at the embankment toe and increase the potential for thaw subsidence there. Further, dust deposition from the road on the adjacent tundra is favoured and can alter snow-related thermal insulation (Raynolds et al.,

2014). These examples underline the need for capturing fine-scale components (meter scale) when modelling infrastructure elements.

Comparable to the challenge of modelling microtopographic landscape features such as for polygonal tundra dynamics (Pau et al., 2014; Kumar et al., 2016; Abolt et al., 2018; Nitzbon et al., 2019; Nitzbon et al., 2020), a key limitation of modelling infrastructure in current LSMs is in their coarse spatial grid, given typical horizontal model resolutions of ~100 km. Therefore, to the best of our knowledge, no studies using LSMs have analysed consequences of permafrost degradation on Arctic infrastructure so far. Other model-based studies of Arctic infrastructure failure are rare and are usually limited to geotechnical site-specific studies (Darrow 2011; Fortier et al., 2011; Flynn et al., 2016; O'Neill and Burn, 2017).

The following sections give a short overview of differing modelling strategies of simulating infrastructure state and permafrost degradation, underlining their strengths and limitations.

**1.2.1    Geotechnical models (GTMs)**

With a focus on fine scales and the aim to describe site-specific conditions and complex processes and interactions, *Geotechnical models* (in the following referred to as **GTM**s) are specifically designed for engineering problems. The temporal focus of the modelling is on the construction phase (months to a couple of years), but does not usually account for longer time horizons. The spatial focus is on capturing thermal distributions in the ground with a 2D or 3D imprint. Boundary conditions are often prescribed as stationary present day temperature and soil moisture state. Prescribing representative boundary conditions for the local site is essential for producing accurate model results (Darrow, 2011). However, model initialization by 2D or 3D temperature and soil moisture fields is usually hampered given the scarcity of measurements, and the computational burden of these models does not allow for long spin-up runs to ensure the model domain is in equilibrium with realistic boundary conditions.

Thermal insulation by snow is a key component of permafrost modelling, but is often not accounted for in GTM studies. More complex *Thermo-Hydro-Mechanical (THMs) models* (e.g. Plaxis, GeoStudio) allow for a coupled model description of thermal, hydrological, and mechanical components, but their high computational costs limit their applicability for long-term climate-warming model runs or parameter uncertainty testing.    T    he model scales are typically limited to a few tens of meters. A strength of GTMs lies in the possibility to couple mechanical processes, and providing specific model diagnostics in the form of stability and deformation measures, failure modes and quantification of the time to failure, which allow inferring direct measures for ground stability and road safety.

**1.2.2    Land surface models (LSMs)**

Representing the land surface components of Earth System Models (ESMs), LSMs describe exchange processes of water and energy fluxes at the land surface-to-atmosphere interface and ultimately enable the feedback from land to the climate system.

Yet, the representation of permafrost within state-of-the-art LSMs contained in the last IPCC assessment was less than
satisfactory (Ciais et al., 2013;Koven et al., 2013), and the latest model simulations submitted for the forthcoming IPCC
assessment have not substantially improved on this (Burke et al., 2020). However, most of the latest IPCC ESM's can at least
realistically account for the insulating effects of snow, and soil water phase changes (Burke et al., 2020). The greatest
shortcoming in the majority of the models is that they lack a sufficiently high vertical resolution in the soil column and a
representation of heat flux in deep soil layers (Burke et al., 2020). The standalone versions of LSM's are generally more
advanced than the ESM versions, due to the additional technical work required for coupling to the atmosphere. In offline mode,
more LSM's account for these key processes, and other major factors such as the impact of organic matter on soil thermal and
hydraulic properties (Lawrence et al., 2012;Chadburn et al., 2015;Kleinen and Brovkin, 2018;Qiu et al., 2019). In addition,
the current development of improved schemes for capturing of water flow dynamics will help in accounting for additional
processes of importance to infrastructure stability, such as ice lens formation or frost heave (Aas et al., 2019). Despite these
efforts, LSMs still miss representing key processes for modelling rapid permafrost degradation, and scale-issues remain a
challenge as the processes involved in abrupt permafrost degradation are very localized.
An established LSM strategy for representing landscape heterogeneity (e.g. for. representing different vegetation types)
consists of    splitting grid cells into "tiles"    to take into account sub-grid variability in surface characteristics    , but such
tiling schemes are not spatially explicit and historically have not described interaction between tiles    (Lee et al., 2014;Ekici
et al., 2019;Cai et al., 2020), while more explicit interacting tiling schemes are in development (Aas et al., 2019). The
combination of these developments will enable a first order quantification of land surface subsidence associated with
permafrost thaw and subsequent land surface transformation such as wetland and thermokarst lake formation.    The benefit
of these developments will be implemented such that the models can help understand and quantify permafrost thaw-associated
feedbacks to the climate system. Such feedbacks can only be systematically quantified using the coupled framework of Earth
System Modelling, which no other models discussed in this study can represent.

### 1.2.3    Process-based Tiling Models (PTMs)

Modelling the interactions between climate warming, permafrost degradation, and infrastructure requires combining the high
level of detail of GTMs with the large-scale and long-term capabilities of LSMs. This challenge of bridging the gap in scales
can be met by the use of computationally-efficient Process-based Tiling Models (PTMs) through reducing complexity while
still resolving key processes for capturing infrastructure-affected permafrost thaw. As opposed to standard LSM "tiling"
approaches as mentioned in the previous section, we here refer to "process-based tiling" by considering the dynamic lateral
interaction between individual tiles. Tiling approaches can be further optimized in efficiency by applying "adaptive tiling"
concepts by using a dynamical number of tiles (Fisher, Koven 2020).
A reduced order representation allows much faster time integration compared to GTMs and therefore allows for adequate
model spin-up, long-term simulations over many decades, and uncertainty quantification through ensemble simulations. The
1-D focus on high vertical resolution of PTM's enables them to resolve time-sensitive processes (such as the build-up of a thin
ice-layer on a pond) and allows them to include complex processes at a scale currently not available in GTMs.
Capturing key dynamic processes despite reduced model complexity is enabled by making use of the flexibility of PTMs in
terms of modularity and scalability to a specific problem. For this purpose a case-specific tiling concept is needed to segregate
infrastructure and tundra into individual structural units adapted to the scales of the modelling task.  Examples of using a PTM
(CryoGrid) in a tiling setting for describing diverse permafrost landscape dynamics can be found in Nitzbon et al. (2019, 2020),
where the authors capture micro-topography features to model dynamics of polygonal tundra degradation, and in (Langer et
al., 2016), where accelerated thaw through talik formation induced by thermokarst lakes is simulated. Jan et al. (2018) used a
mixed-dimensional model structure for efficiently simulating surface/subsurface thermal hydrology in low-relief permafrost
regions at watershed scales. Using the GIPL model, Daanen et al. (2011) linked  large scale climate and permafrost simulations
to small scale engineering aspects with a focus on Greenland.
In this study, we used a laterally-coupled version of the 1D heat conduction PTM CryoGrid3 for modelling impacts from
permafrost degradation caused by climate change on Arctic infrastructure. To demonstrate how permafrost-affected
infrastructure failure though climate change can be captured in a modelling context, we present a case study focusing on gravel
roads built over ice-rich permafrost, particularly the area around the Prudhoe Bay (Dalton highway, Alaska). In section 2, we
give details about our simulation set-up. In section 3, we show model results of permafrost degradation affected by linear
infrastructure for historical, present day and future warming climate conditions. In section 4 we discuss how our modelling
approach can be applied at the pan-arctic scale    where we suggest a strategy for modelling impacts of permafrost degradation
on Arctic infrastructure by bridging the gap between small and large scales. Conclusions are drawn in section 5. Details of
model set-up and model evaluation are given in the appendix.
**2    Model description and simulation setting**
We aimed to simulate the temperature regime under a typical gravel road built on continuous permafrost and its thermal
evolution under climate warming. Given the abundance of infrastructure in the Prudhoe Bay Area (Alaska, coastal plain), we
chose this region for our case study (Fig. 2).

We used a laterally-coupled one-dimensional heat conduction model (*CryoGrid3*) which is able to simulate dynamic surface
and subsurface processes in permafrost landscapes (Langer et al., 2016;Westermann et al., 2016). The model explicitly
simulates ground subsidence and subsequent ponding as a consequence of melting of excess ice in the ground (Westermann et
al., 2016). The CryoGrid3 model includes a dynamic snow cover representation and an explicit surface energy balance scheme
for simulating heat and water exchange with the atmosphere. Our model version also accounted for lateral heat and water
fluxes in the ground between adjacent tiles (Nitzbon et al., 2019). Vertical water flow is described by an instantaneous

infiltration scheme, while the calculation of lateral water fluxes between adjacent tiles is based on Darcy's law, assuming a constant lateral hydraulic conductivity. The model has been extensively evaluated with respect to its performance in representing key physical processes in tundra landscapes, namely surface energy balance, thermal state of ground, soil hydrology, and snow cover dynamics (Westermann et al., 2011;Langer et al., 2013;Westermann et al., 2016), including subsurface water exchange (Nitzbon et al., 2019;Nitzbon et al., 2020). Further, the model has been evaluated under different climate conditions and for different permafrost landscape types such as peat plateaus and palsas (Martin et al., 2019).

## 2.1 Model limitations

Although our model was designed to capture essential components of permafrost thaw, it is limited by several physical assumptions.We do not model heat advection by lateral water flow which can strongly accelerate road failure (de Grandpre et al., 2012;Chen et al., 2019). Our model represents the thermal impact of the snow cover on the ground by a simple bulk snow density approach neglecting any temporal changes in snow cover properties resulting from snow metamorphism (i.e., depth hoar) or wind compaction (i.e., ice crusts). However, the effective thermal properties of the snow layer can change following infiltration and refreezing of rain or meltwater. The snow scheme emulates lateral redistribution of snow, but without explicitly calculating wind drift. Besides infrastructure failure due to thawing of ice-rich soils, frost heave in frost-susceptible ground below the embankment can also lead to severe infrastructure damage - a process which is not captured in our model.

In this study we focus on ice-rich grounds and assess the timing when infrastructure failure might occur in the future due to melting of excess ground ice and subsequent subsidence. We do not model mechanical deformations of and stresses in the embankment which can largely determine ultimate road stability especially in areas of lower ground ice contents. The exact timing of failure will depend on road design (e.g. embankment dimensions, Kong et al, 2019) and local climate conditions which can of course deviate from the case that we are discussing here (especially if road protecting cooling devices are installed such as thermosiphons, ducts, insulating layers, or snow sheds). Rather than trying to simulate a specific infrastructure element at a given location and inferring specific estimates of infrastructure failure, we want to demonstrate the broader applicability of our modelling approach for capturing critical processes of key importance to the stability of infrastructure on ice-rich permafrost grounds which are not represented in other modelling frameworks.

## 2.2 Model set-up

Down-scaled climate data (SNAP (SCENARIOS NETWORK FOR ALASKA and ARCTIC PLANNING), Lader et al., 2017) was used to force our model under historical, present day and projected climate change, covering the period from 1975 to 2100. We bias-corrected climate forcings to ensure that our simulated present day soil temperature conditions were close to observations (see appendix). For describing future climate evolution we used the RCP8.5 scenario, which leads to mean annual surface air temperature warming of about 5° C above year 2020 levels at Deadhorse at mid-century.

We set up our model to represent a transect from the road centreline to the adjacent tundra up to a distance of 100 m. The
tundra and infrastructure elements were accounted for by prescribing different ground stratigraphies (Tab. 1 and Fig. 3), and
by adapting surface- and below-ground specific parameters (Tab. 2). We investigated a 'conservative' and a 'vulnerable' case
for our model simulations. In the latter case we consider pond formation next to the road and increased solar incoming radiation
on the road shoulder by assuming southerly facing conditions. Pond formation has a pronounced effect on our modelled soil
temperatures by altering surface energy fluxes through lowering surface albedo and replacing thermal properties of the soil
surface by those of a water body. Further we assumed snow-free conditions on the road surface and additional snow
accumulation on the road shoulder and toe (Fig. 3). Snow accumulation and ponding next to the highway are primary drivers
of thaw in our model-setup. Further drivers such as dust deposition, lateral heat advection by subsurface water transport or
backing up of surface flow are not accounted for.
We used a high vertical resolution for grid cells in the upper 4 meter of the ground and coarse resolution towards our lower
model boundary at 1000 m depth, which is subject to an assumed geothermal heat flow of 0.05 W m$^{-2}$ (Lachenbruch et al.,
1982). In the appendix we give detailed information about model spin-up, ground stratigraphy choices, and gravel road design
choices. Table 2 lists key model parameters which affect surface and subsurface heat fluxes. Further standard model parameter
choices can be found in Westermann et al. (2016) and Nitzbon et al. (2019).

## 2.3    Simulation setting

The model domain was subdivided into structural units (SUs) of differing surface and subsurface characteristics (e.g., with
respect to albedo, snow accumulation, excess ice presence and ground composition, see Fig. 3). All SUs were laterally
connected to allow for below-ground lateral exchange of heat and water between the gravel road and the adjacent tundra. The
sensitivity of model results to horizontal model resolution was investigated using three set-ups to capture the gravel road and
tundra up to a distance of 100 m from the road centre with increasing horizontal resolution (Tab. 3 and appendix).

## 3    Results and discussion

### 3.1    Present day conditions

Figure 4 illustrates how a replacement of the tundra surface by a gravel embankment is affecting subsurface temperatures: the
removal of the protective peat layer and the clearance of snow on the road centre increases the coupling of surface air and
ground temperatures. Warm summer air temperatures penetrate much deeper into the ground under the road surface as
compared to the tundra (Fig. 4, right panels). Snow accumulation at the toe and road shoulder results in markedly warmer
winter ground temperatures (Fig. 4, left panels).

Our simulation results describe a gravel road which is stable under year 2000 climate conditions as we model an artificial permafrost table under the road which is elevated above the tundra permafrost table    (Fig. 4). Warming from the beginning of the century until today (year 2020) has increased overall ground temperatures, which leads to a pronounced active layer deepening under the toe (SU3). In this area, snow accumulation and related soil warming leads to excess ice melt and pond formation in the year 2014, further enhancing subsoil temperatures. Our model results are in line with our observations taken along a transect away from the Dalton highway close to Deadhorse (see appendix), which show increasing winter ground surface temperatures with declining distance to the road (Fig.A2), underlining the thermal imprint of snow accumulation at the shoulder and toe. A maximum in thaw depth under the toe of a gravel road describes a characteristic also inferred in GTM studies and observations (Flynn et al., 2016,  Fortier et al., 2011,  Qi et al, 2012).

Under present day climate conditions (year 2020), the active layer at the toe is close to reaching the depth of the embankment base - in contrast to the active layer below the road centre which is still elevated relative to the toe. The active layer in the tundra (SU4) is around 70 cm by 2020, in line with observations (CALM Database. Washington, D.C.: The George Washington University, https://www2.gwu.edu/~calm/data/data-links.html.). The formation of a zone of temperatures approaching zero degrees at the end of the winter season (light bluish areas in Fig. 4, March 2020) indicates vulnerable conditions where additional future warming will have a strong effect on thaw depths. The general characteristic of strongly increased heat penetration below the road centre as compared to the tundra and the increased soil warming under the toe caused by snow accumulation was also inferred in other modelling approaches (Fortier et al., 2011;Flynn et al., 2016, Darrow 2011). For a vegetated hillslope side, Jafarov et al. (2018) inferred a pronounced effect on soil warming from preferential snow accumulation.

For investigating the impact of infrastructure on permafrost degradation we have additionally run a reference run (single tile) with identical parameter choices, but without including infrastructure. It can be seen that the outer tundra tiles from the laterally coupled setting reach soil temperatures very close to the reference run (separate right soil temperature columns), therefore justifying our assumption of a zero boundary flux at the outer model domain.

### 3.2    Development under future climate change

Under the 21$^{st}$ century future climate change scenario, the simulated ground thermal regime shows strong increases in maximum thaw depths. The prescribed additional mean annual surface air warming of about 5 °C above 2020 levels (7 °C above 2000 levels) until the mid-century for the Prudhoe Bay region (following RCP8.5) leads to the formation of an open talik below the pond (toe tile, see discussion below). This warming signal further spreads into the embankment, subgrade and adjacent tundra (Fig. 4, see also the animated temperature evolution in the supplementary material). The thermal buffering of the lower shoulder is not sufficient anymore to prevent strong thawing in the embankment subgrade. In the case of the thaw

front reaching thaw-unstable ice-rich layers, subsequent ground subsidence would result in failure of the embankment shoulder. In contrast, the snow-free road centreline is stable by 2050 as winter heat extraction is still strong enough to prevent active layer deepening below the embankment base. Nonetheless, further warming and lateral growth of the talik into the embankment core and subgrade will strongly destabilize the road foundation and lead to subsequent total road failure if the road was built on ice-rich ground (Fig. 4 lowest panels).

Our results under the 21st century climate change scenario reveal pronounced differences in the dynamics of simulated thaw depths among the different structural units (Fig. 5; road centre, shoulder, toe, tundra). Thawing rates can be split into two periods: a period of slow and gradual increase in maximum thaw depths, followed by a sharp increase in thawing rates once a critical warming of the ground has been reached. The two-phase thawing characteristic cannot be explained through changes in external climate forcing (see Figure A3), but by internal dynamics following the formation of year-round partially unfrozen layers in the ground. We can roughly track the timing of the increase in thawing rates by the first occurrence of a layer with a maximum refreeze of 25% of the pore water (dashed vertical lines, Figure 5). This diagnostic can be seen as a precursor for later full talik formation (i.e. year-round unfrozen conditions). The occurrence of individual cold or snow-poor winters can result in a strong winter refreeze of taliks, causing large year-to-year variability in simulated maximum thaw depths.

As a consequence of snow accumulation and subsequent ground warming, the toe structural unit is the first location where an increase in thawing rates is seen. In the 'vulnerable' setting with ponding at the toe, this increase is very pronounced and starts already around 2015 after pond formation (Fig. 5b), reaching thawing rates of 20 cm yr$^{-1}$. Such high thawing rates are an expression of the heat gain after the replacement of the snow-free tundra surface (assumed albedo of 0.2) with a water surface albedo (0.07, see table 2). Further, an unfrozen pond allows for a more efficient heat uptake of warm surface air through vertical mixing within the water column as compared to top-down heat diffusion in solid ground. The shoulder tile reveals a lagged thaw rate increase, reaching thaw penetration below the embankment base around 2040 (around 2055 if no pond is assumed), which corresponds roughly to the formation time of a year-round partially unfrozen soil layer (Figure 5). Further warming will ultimately lead to road destabilization if the ground below the embankment base is ice-rich. In contrast, the road centre is much more stable with a thaw rate increase many decades later. The outer road edge is destabilized around the year 2060 (2075 without ponding). Talik formation under the road centre only occurs after year 2080, but triggers a very pronounced increase in thawing rates (Fig. 5). The large increase in late-century thawing rates under the road centre are a consequence of previous continuous warming of subgrade temperatures through lateral heat flux in depth. This heat flux leads to an almost isothermal temperature depth profile under the road close to 0 °C (see Fig.4, lowest panels), making the ground strongly vulnerable to further warming. In contrast to the tiles which are thermally affected by the gravel road, the adjacent tundra reveals only a gradual active layer deepening without the formation of largely unfrozen layers or full taliks and with no accelerated thaw in the 21$^{st}$ century (Figure 5).

Our inferred dynamics of thaw rate acceleration resemble the thawing behaviour seen in a model study of a gravel road on discontinuous permafrost (Fortier et al., 2011). Figure 5 underlines that model studies that do not explicitly incorporate infrastructure in their model design are likely to strongly underestimate the timing of infrastructure damage following permafrost thaw, as infrastructure presence can promote enhanced ground warming.

We further tested how a stabilization of climate at the end of the 21st century would affect the long-term behaviour of ground thaw dynamics. All tiles which are affected by the presence of the gravel road show continuously increasing thaw depths throughout the 22$^{nd}$ century as deep soil layers reach above-zero temperatures. In contrast, the active layer below the tundra stabilizes at depth (Figure A3). This stabilization is realized despite slightly positive mean annual air temperatures after year 2075 (Figure A3, light blue curves) through combined effects from a pronounced reduction in snow insulation (as a consequence of strongly reduced snow heights and shortened snow seasons) and soil surface drying during summer (resulting in a strong increase in summer insulation of the soil surface). This finding shows the key impact of the protective peat layer for sub-ground temperatures and permafrost state, but we underline that our simulated preservation of tundra permafrost also depends on the chosen model setting (e.g. with respect to external climate forcing and to internal model parameterizations).

## 3.3   Sensitivity to number of structural units

A key question of interest to modelling concerns the amount of detail necessary to satisfactorily capture the road thermal state and its adjacent environment. We investigated this aspect by running our model with low lateral resolution (1 road tile, 1 tundra tile), with medium resolution resolving all structural units (road, shoulder, toe, tundra) and with high resolution. In the latter case we further subdivided structural units into multiple tiles (total of 30 tiles, Tab. 1).

When describing our model-setup by only two tiles, our results suggest that permafrost degradation under the road centre is underestimated, resulting in less pronounced thaw  The medium and high resolution settings show consistent results, with slight differences when focusing on the outer edge of the road (Fig. 6a). In contrast, modelled maximum thaw depths for the adjacent tundra show no sensitivity to horizontal model resolution (Fig. 6b). The toe and shoulder tiles (Fig. 6c, d) are not resolved in the low resolution setting and reveal much stronger permafrost degradation, underlining the importance of accounting for these elements in the model design. While the difference in simulated thaw trajectories between medium and high resolution runs is rather small for the shoulder tile (Fig. 6c), larger deviations are seen for the toe tile (Fig. 6d). Yet both experiments reveal very similar thaw dynamics and reveal differences in the timing of thaw below the embankment base of only a few years.

## 4    Perspective on future modelling of Arctic infrastructure failure

Our results clearly exhibit that there are combined effects from climate warming, permafrost degradation, and the presence of infrastructure. This highlights that future model-based risk-assessments of Arctic infrastructure should take into account all three components in understanding and quantifying the consequences of permafrost degradation on Arctic infrastructure. For such a purpose the use of numerical modelling appears the most appropriate tool to account for the full spectrum of dynamical permafrost changes expected in a warmer future.

Despite the large difference in scales between GTMs (decimetres to meters) and LSMs (hundred kilometres), our study illustrates a potential for bridging this gap by the use of Process-based tiling models (PTMs). PTMs are computationally less expensive than complex three-dimensional GTMs, structurally more flexible in describing processes relevant to permafrost degradation than both GTMs and LSMs, and adaptable to different scales. The computational efficiency of PTMs allow them to conduct long-term climate change scenario simulations while also exploring uncertainty by simulating ensembles.

In light of risk assessments of Arctic infrastructure failure, we envision two key directions of model applications: I) large-scale (i.e. LSM scale) pan-Arctic modelling of climate-induced infrastructure failure for inferring potential risk estimates, and II) fine-scale (i.e. GTM scale) modelling at individual infrastructure level to determine specific risk estimates. While large-scale modelling aims at a rather general quantification of how infrastructure could potentially be affected through climate change in certain regions across the Arctic, fine-scale modelling allows a case-specific assessment of certain infrastructures at chosen locations. PTMs can support both modelling approaches.

### A) Linking PTMs to LSMs/ESMs

Given the focus on large-scale risk assessments, LSM development aims at coarsely capturing general aspects of climate-change related infrastructure risks. This could be realized by implementing infrastructure through tile-based infrastructure classes, with each class representing a certain type of infrastructure (e.g. roads, pipelines, buildings). The result of LSM simulations representing infrastructure in such a reduced but therefore manageable manner (Cai et al., 2020) could produce pan-Arctic risk maps showing key regions where infrastructure failure through melting of excess ice and subsequent ground subsidence is likely for a given infrastructure class under a given scenario of future climate change.

We envision three different directions of how PTMs can be linked to LSMs for modelling of large-scale infrastructure risks from permafrost degradation:

(I) Drawing general conclusions from PTM analyses can support LSM development in terms of how to best incorporate tiling-based infrastructure descriptions into LSMs. For this purpose, PTMs could e.g. inform about the minimum number of structural units needed to satisfactorily simulate the interaction of a certain infrastructure type with its permafrost surrounding. Here, we showed that for a gravel road setup, four structural units (i.e. four tiles) are a sufficient

approximation to describe the impacts of snow accumulation and ponding on the thermal state of the road. Other infrastructure elements will require different numbers of structural units depending on infrastructure design.

(II) Rather than developing tiling-based infrastructure descriptions in the LSMs themselves, PTMs with an infrastructure component could be run offline for each LSM grid cell. This allows a quantification of pan-Arctic infrastructure risks based on the climate forcing and soil grid information (such as assumed excess ice distribution, Cai et al 2020) from a LSM, in combination with inferred permafrost-infrastructure interactions as calculated by the PTM.

III) A third direction (as done in this study) is the direct use of climatological forcing data from an ESM to run a PTM in a standalone mode in which site specific aspects can be accounted for (such as local soil stratigraphy, soil moisture, ground ice distribution).

Pan-Arctic large-scale modelling as discussed above (I and II) can only provide a potential risk estimate, as they do not explicitly simulate the stability of individual infrastructure at specific locations. Site specific risk assessments must be inferred from complex three-dimensional GTM simulations which account for spatial details and therefore are capable of considering site and design-specific infrastructure aspects (such as infrastructure dimensions (Kong et al., 2019), asymmetry (Raynolds et al.,2014, Abolt et al.,2017), presence of cooling measures (Xu and Goering, 2008), and local boundary conditions (e.g. local climate, insolation angle, subgrade hydrological flow, Darrow 2011).

**B) Linking PTMs to GTMs**

PTMs could improve GTM setups by providing information about climate-induced changes in upper model boundary conditions. GTMs often assume that the upper model boundary can be described by quasi-stationary conditions concerning atmosphere-to-ground heat fluxes (e.g. by using fixed n-factors). Darrow (2011) underlines the high sensitivity of modelled ground temperatures to upper boundary assumptions and points to limitations of using fixed n-factors when climate change effects are to be considered. In our study we show the strong enhancement effect of permafrost degradation coming from ground heat gain caused by ponding next to the road. Such effects are typically not resolved by GTM models. A model-based representation of pond formation or of snow cover changes require high vertical resolution and small time steps. Although this could in principle be represented in GTMs, the computational demand for running such models on climate timescales in very high resolution in 2D or 3D is still beyond reasonable computational effort today. In contrast, the 1D focus of PTMs on capturing vertical processes in high resolution (and with some 2 or 3D interaction) allows for investigation of complex environmental changes (and their lateral interactions) in a schematic way with moderate computational costs. This can help in drawing more general conclusions on permafrost-infrastructure interactions (under present climate and under climate change). Here we sketch three possible directions of how GTM development could profit from PTM capacities:

(I) PTMs can be used to estimate how strongly changes in ground surface processes affect n-factors. This information could then be used for improving GTM model setups (by re-scaling n-factors) when climate change effects are to be considered.

II) A further improved representation of upper boundary conditions in GTMs could be reached by hybrid model approaches which take advantage of the individual strengths of different model classes. One strategy could be the coupling of the complex surface energy balance component of a PTM to a GTM for use as upper model boundary forcing ("semi-hybrid approach").

(III) Such a modelling approach could be extended into a "full hybrid approach" by running the PTM on a comparable horizontal resolution as used in a GTM. The simulated thermal state from the PTM could then be applied to the GTM for calculating soil hydrology and mechanics. In turn, GTM information is returned to the PTM, such that the state of each model is updated by the results of the other in each iteration. While losing some GTMs resolution of geometries in the thermal field calculations, such an approach would benefit from the complex process description and modular adaptability of the PTM, to better describe some aspects of infrastructure degradation.

The best-suited modelling strategy will finally depend on the type of risk analysis under investigation. In the light of degrading permafrost observed already today in many regions around the Arctic, novel modelling strategies for estimating risks of future infrastructure failure under a warming climate are urgently needed.

## 5    Conclusions

In our study we show the applicability of a laterally-coupled one-dimensional heat conduction model for simulating permafrost-affected infrastructure failure, demonstrated for the case of a gravel road built on continuous permafrost. Our simulation results allow an improved process understanding of how the ground below infrastructure can degrade under a scenario of intensive future climate change, and underline the potential for use of such models in pan-arctic risk assessments.

Our model simulations show a transition from slow and gradual thaw to fast, and likely irreversible permafrost degradation, and point to a threshold-like behaviour where (without extensive active ground cooling measures) road failure is inevitable once a critical level of ground warming has been reached. Rather than a simple top-down thawing and deepening of the active layer below the road centre, we identify lateral destabilization of the embankment and subgrade caused by talik formation under the toe, a key process which results in accelerating thaw rates and results in subsequent road failure.

The comparison of modelled thawing depths under the road and tundra underlines that infrastructure can exert a strong impact on ground temperatures as a consequence of increased net heat input in the subsurface. Therefore it is crucial that model-based estimates of the timing of Arctic infrastructure failure account for the amplification of permafrost degradation through the presence of Arctic infrastructure itself.

Based on our modelling results with differing lateral model resolutions we conclude that the minimum number of model tiles should be chosen such that important *structural units* are captured, such that small-scale processes which exert a key impact on the ground thermal regime are accounted for. In our example of a gravel road we have identified snow accumulation and ponding at the shoulder and toe such key processes. For our linear infrastructure example we suggest a minimum number of four structural units to represent the road surface, the shoulder, the toe, and the tundra. For other types of infrastructure, structural units have to be adapted accordingly. If more spatial detail is needed (such as the information about the thermal state of the embankment at the road edge), structural units can be further subdivided into individual tiles of finer lateral resolution. Our simulations demonstrate that a low number of structural units can be sufficient to resolve processes operating on engineering scales. Our use of a laterally-coupled one-dimensional model allows us to infer results in line with more complex 2D- and 3D-models but our model's computational efficiency enables us to calculate long-term climate change impacts on infrastructure from permafrost degradation. Further, our simulation results show the potential for reducing model complexity and therefore underline the capability of use in computationally extensive pan-arctic analyses.

Our results underline that it is crucial to consider climate change effects when planning and constructing infrastructure on permafrost as a transition from a stable to a highly unstable state can well occur within the infrastructure's service lifetime (about 30 years). With our focus on the Dalton highway at Deadhorse (Prudhoe Bay, Alaska), we illustrate that such a transition can even occur in the coming decade for infrastructure built on continuous permafrost that displays cold and relatively stable conditions today.

## Appendix A:     Details of model set-up

### A1     Climate Forcing

We used down-scaled climate data (for air temperature, humidity, pressure, wind speed, rain and snow precipitation, and incoming long-wave and short-wave radiation) provided by the SNAP database (SNAP, (Lader et al., 2017) to force our model under historical, present day and projected future climate change conditions. We follow the approach by Westermann et al. (2016) of using monthly climate anomaly fields in combination with high-frequency reference climate data to generate bias-corrected climate forcings. The monthly anomalies are calculated based on a downscaled GFDL model run under the RCP8.5 scenario. The reference climate data cover the period 2005 to 2015 and are taken from the downscaled SNAP ERA data. The RCP8.5 forcing describes a scenario of extensive future climate change resulting in a radiative forcing of 8.5 W m$^{-2}$ by 2100. For the Prudhoe Bay area (Deadhorse) this forcing leads to mean annual air temperatures about 5° C above mean year 2020 levels by the year 2050 (about 8° C by 2075).

## A2    Initialization and boundary conditions

We initiate soil temperatures based on borehole data at 10 m and 20 m depth (Romanovsky et al, 2019) and start our simulations in the year 1975, using the first 25 years for spinning-up our model domain under historical climate conditions.          We prescribe an external water flux of 2 mm day$^{-1}$ for the tundra and toe tiles for the period of unfrozen soil surface conditions. This flux could mimic the impact of surficial lateral water fluxes or could be understood as a correction factor for precipitation biases. We introduce this flux for capturing observed high soil moisture conditions in tundra soils next to the road at our chosen location.

Based on borehole data (Raynolds et al., 2014), we prescribe the tundra soil stratigraphy by assuming a 30 cm peat layer on top of a two meter thick silty mineral soil layer. Below, a sandy mineral soil layer is assumed extending to the bedrock boundary at 10 m depth (see Fig. 3). In our model setting with ponding ('vulnerable' setting), we assume a layer of high excess ice between 1 m and 2 m, mimicking a case of a buried ice wedge next to the road. The maximum model depth is constrained to 1000 m with a lower boundary flux condition given by a geothermal heat flow of 0.05 W m$^{-2}$ (Lachenbruch et al., 1982). Ground stratigraphies for tundra and infrastructure conditions are given in Tab. 1.

In our model setup we consider a transect from the road centreline to the adjacent tundra up to a distance of 60 m (low and medium resolution) or 100 m (high resolution), assuming symmetry along the centreline for computational efficiency. At the outer model boundaries we assume zero lateral flux conditions given our symmetry assumption with regard to the road centreline and negligible lateral gradients at the outer tundra model boundary.

## A3    Model resolution and subdivision into structural units

We have run CryoGrid3 in three different set-ups in which we have captured the gravel road and tundra up to a distance of 100 m from the road centre with increasing horizontal resolution. In a low resolution setting (*LowRes*) we have only used two tiles (i.e. simulating two connected soil columns) to describe the gravel road centre by one SU coupled to a tundra SU. In contrast, in the medium resolution (*MidRes*) and high resolution (*HighRes*) settings we use four SUs which allow us to resolve our model domain in greater spatial detail, e.g. by accounting for small-scale effects of snow accumulation and ponding in the vicinity of the road (see Fig. 3). In these settings we additionally consider one structural unit for the embankment shoulder (SU2) and one structural unit for the toe (SU3). Further, we resolve the outer edge of the road surface separately by one additional tile (*MidRes*). In the *HighRes* setting we describe the tundra with finer horizontal resolution close to the road, and increasingly coarser resolution towards the outer boundary at 100 m distance from the road (Tab. 2).
The vertical model grid node spacing is 2 cm in the upper 4 m, 10 cm between 4 m to 10 m, 20 cm in the depth range 10 m to 20 m, followed by 10 more layers of increasing thickness down to the lower model domain boundary at 1000 m.

533

**A4    Road embankment and toe**

We assume that the road surface is 2.5 m above ground with a total embankment thickness of four meters (i.e. we imply an excavation of the uppermost 1.5 m of ground during construction). The material in the top 10 cm of the road surface is fine-grained gravel, all other parts of the embankment are assumed to consist of coarse-grained gravel. We assume that the road centre is permanently snow free as a consequence of snow plowing. We capture this effect in the model by removing snowfall from the road centre.

We constrain landscape-scale maximum snow height on the adjacent tundra to 40 cm to simulate snow heights in the range of observations (Nicolsky et al., 2017). As we do not simulate snow redistribution (through plowing and wind drift), we realize additional snow accumulation at the toe and shoulder by scaling snowfall by a factor of four. We constrain total snow heights depending on the distance to the surface road edge and describe a linear profile which results in largest snow heights at the toe (see Fig. 3, light blue shading). In our *LowRes* setting we cannot resolve ponding or snow accumulation along the shoulder and toe, as we only include one road tile and one larger-scale tundra tile with uniform snow height.

The embankment has an assumed slope of 1:2 (1 m vertical vs. 2 m horizontal) and determines the maximum height of snow accumulation at the shoulder and toe (Fig. 3). In our conservative setting, we assume a general orientation of the Dalton highway in North-South direction. In our vulnerable setting we assume a southerly facing road shoulder and account for increased incident solar radiation. We acknowledge that a specific gravel road can deviate strongly from our assumed setting here (e.g. (Andersland and Ladanyi, 1994).

**Appendix B:  Evaluation of the impact of snow accumulation on soil temperatures**

During the winter season 2018/2019 we have measured soil surface temperatures along a transect at the Dalton highway 10 km south of Deadhorse (70.099°N, 148.511°W, https://doi.org/10.1594/PANGAEA.914327). We have used iButton temperature loggers (model "DS1921G") placed in the uppermost soil surface, covering both sides of the road up to a distance of 50 meters away from the road centre (see Fig. A1). Our transect is in the direct vicinity of the weather monitoring station DSS1 (Dalton MP 405) of the Water and Environmental Research Center (WERC) of the UAF (Toniolo et. al., 2020) which we used for comparing measured soil temperatures to observed surface air temperatures.

For investigating the impact of ground warming from snow accumulation at the road shoulder and toe, we have analysed the temperature difference between the soil surface and the surface air (2 m height) for snow covered conditions (1st of November to 30st of April). The observations show warmer soil surface winter temperatures in the vicinity of the road (SU2 and SU3) compared to the more distant tundra (SU4, Fig.A2, upper panel). In contrast, the snow-free road surface (SU1) reveals mean modelled temperatures about 10 degrees colder than observed tundra soil surface temperatures. By prescribing snow

accumulation at the shoulder and toe (SU2 and SU3), our model simulations capture the snow warming effect and point to a strong year-to-year variability of the magnitude of this effect depending on climatic conditions of a specific year. For the snow season 2018/2019 our observations suggest an additional soil surface warming from snow accumulation of about 4 °C (Fig.A2, lower panel).

**Code and data availability**

The model source code used for the simulations in this work is available at Github (https://github.com/CryoGrid/CryoGrid3/tree/xice_mpi_IS) and will be finally archived on Zenodo in case of final acceptance of this manuscript. The measurement data from Langer et al. (2020) used for evaluation of the model results are available from PANGAEA (https://doi.org/10.1594/PANGAEA.914327).

**Video supplement**

The supplement to this article (https://doi.org/10.5446/47699) contains an animated video showing the results of the simulations described in section 3.

**Author contributions**

TSvD and ML designed the study. TSvD extended the model code for including infrastructure, carried out the simulations and wrote the paper. SJ generated model forcing data sets. All authors interpreted the results and contributed to the paper.

**Competing interests.**

All authors declare that they have no conflict of interest.

**Acknowledgements**

This work was supported by the Federal Ministry of Education and Research (BMBF) of Germany through a grant to Moritz Langer (no.01LN1709A). Additional support came from the AWI Innovation Funds (Innovation Project IP10200006). Special thanks to Alexander Oehme for providing climate forcing data, and to Heiko Gericke and Natalja Rakowsky for IT support. Donald A. Walker acknowledges support by U.S. NSF awards 1263854 and 1928237 and Vladimir Romanovsky acknowledges support by the National Science Foundation (Grant 1832238) and by the Next-Generation Ecosystem Experiment (NGEE-Arctic) project of DOE. This work was also supported by the HORIZON2020 (BG–2017–1) project Nunataryuk (Grant agreement no. 773421).

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

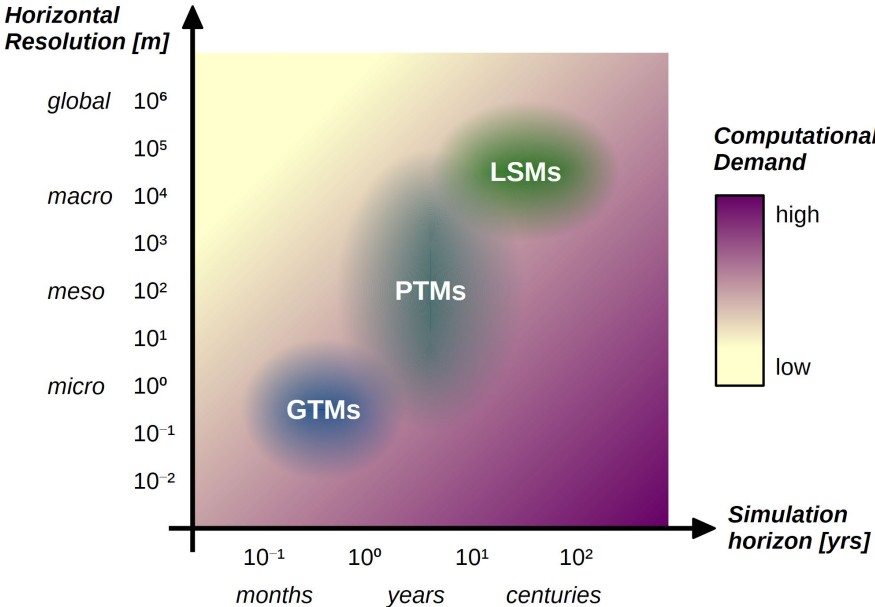



**Figure 1: Classification of model classes according to their representative temporal and spatial scales. Geotechnical models (GTMs)**
**operate on short time horizons (with a focus on the construction period), Land Surface Models (LSMs) focus rather on decadal to**
**multi-centennial timescales. Process-based Tiling models can fill the gap in scales between GTMs and LSMs.**


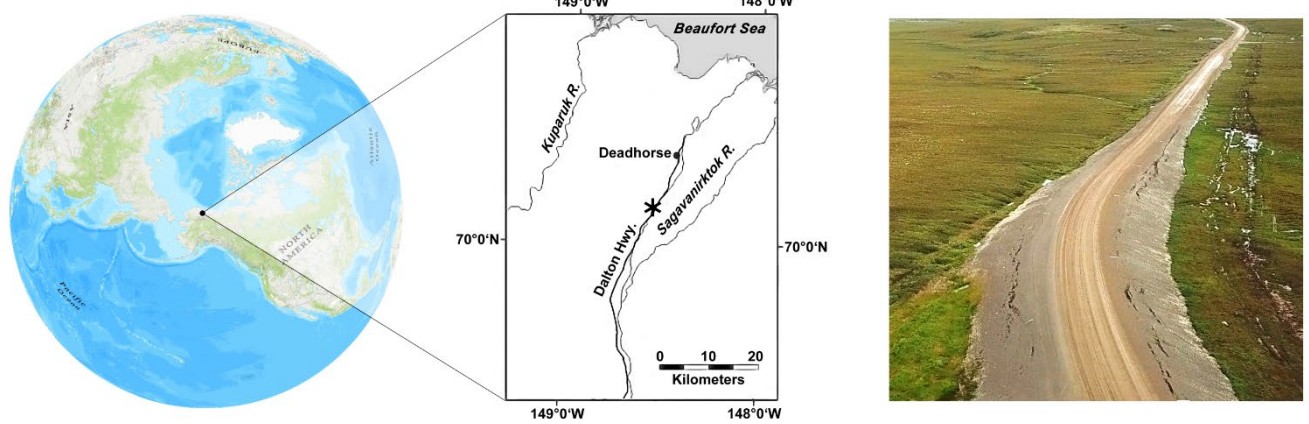

Map source: Esri, HERE, Garmin, Intermap, increment P Corp., GEBCO, USGS, FAO, NPS, NRCAN, GeoBase, IGN,
Kadaster NL, Ordnance Survey, Esri Japan, METI, Esri China (Hong Kong), © OpenStreetMap contributors 2020. Distributed under a Creative Commons BY-SA License.


**Figure 2: Focus of the model domain in the Prudhoe Bay region, Alaska (Deadhorse, 70.099°N 148.511°W, middle figure). Exemplary**
**drone view of the Dalton highway close to Toolik (right figure). The asterisk in the middle figure illustrates the location of our soil**
**surface temperature monitoring site (see appendix).**



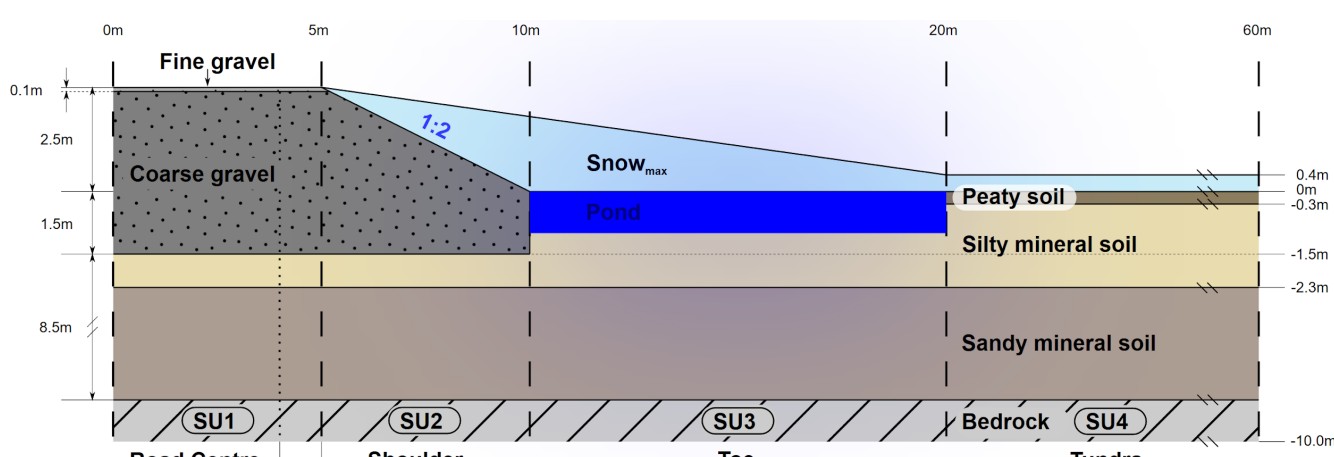


**Figure 3: Modelled half-cross-section from the road centre to the adjacent tundra. The setting shown illustrates the sub-division into 4 structural units (SU1-4: Road Centre (including the additional tile for resolving the outer edge), Shoulder, Toe, Tundra). The greyish area with black dots represents the road embankment. The light blue shading indicates potential maximum snow height. The dark blue area illustrates ponding next to the road. The lower model boundary (not shown) is at 1000m depth.**

806

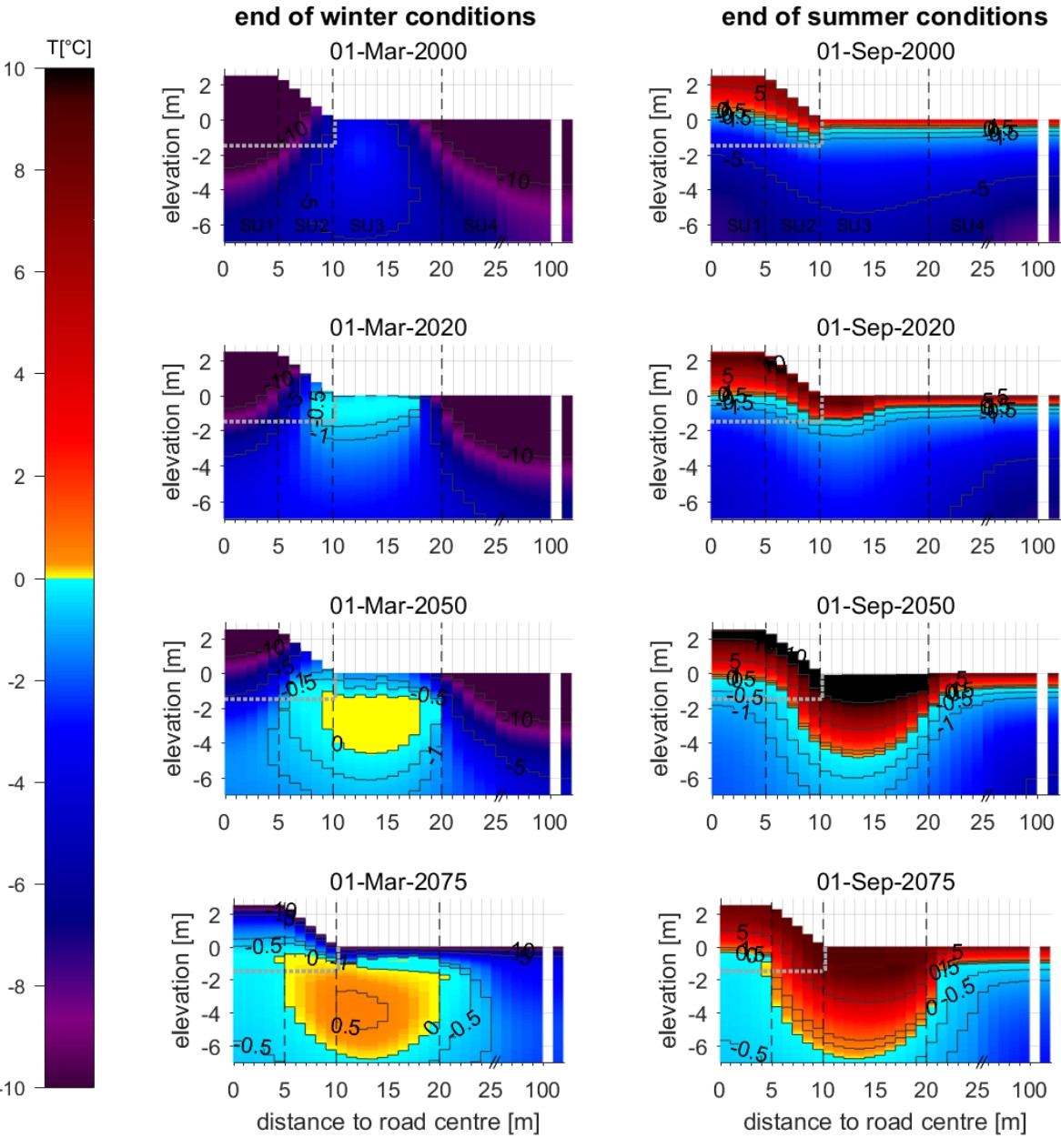

807

**Figure 4: Simulated ground temperatures in the road embankment, subgrade and adjacent tundra at the end of winter (left) and summer (right) for past (uppermost panels), present-day (upper mid-panel), and future (lower 2 panels) climate conditions (based on the RCP8.5 scenario). Results are inferred from the vulnerable *highRes* model setting using 30 tiles. The separate right columns indicate the temperature profile of a single-tile undisturbed tundra reference run. Dotted grey lines illustrate the embankment base. Vertical dashed black lines indicate the separation into the structural units 'road surface', 'shoulder', toe', and 'tundra'. Note the x-scale break for distances larger than 25 m. The lower model boundary is at 1000 m depth (not shown). The position of the 0 °C isotherm (separating bluish and yellowish colours) at the end of the summer corresponds approximately to the maximum annual thaw depth (right column).**

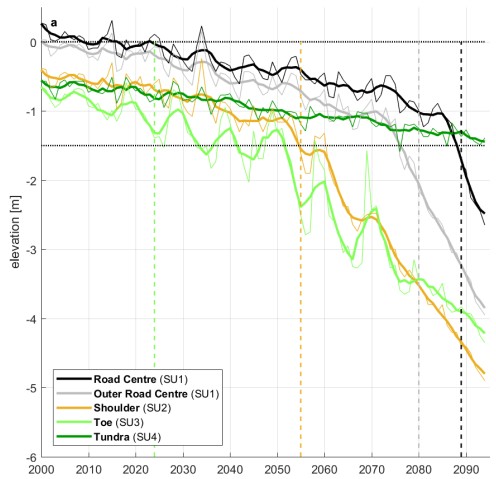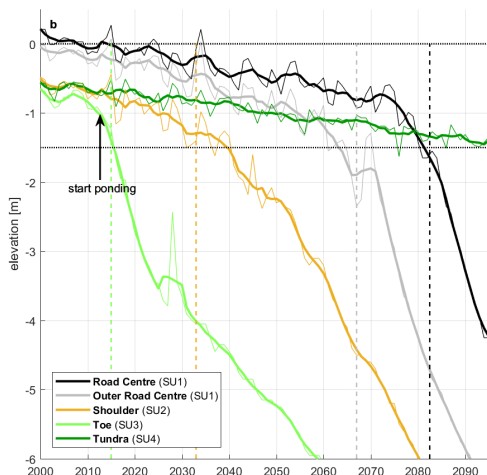

817

**Figure 5: Temporal evolution of maximum thaw depth (MTD) for all structural units under RCP8.5 warming (SU1: road centre, SU2: embankment shoulder, SU3: toe, SU4: adjacent tundra).     Panel (a) illustrates the conservative case without ponding at the toe, panel (b) shows the vulnerable case with ponding and a southern facing road shoulder. Runs show results from the *MedRes* setting with 5 tiles (see table 3) and show annual means (thin lines) and 5 year moving means (thick lines).  Dashed vertical lines indicate the timing of the first occurrence of a year-round partially unfrozen layer (maximum refreeze up to 25% of pore water). The horizontal black dotted lines indicate the tundra surface and embankment base.**

824

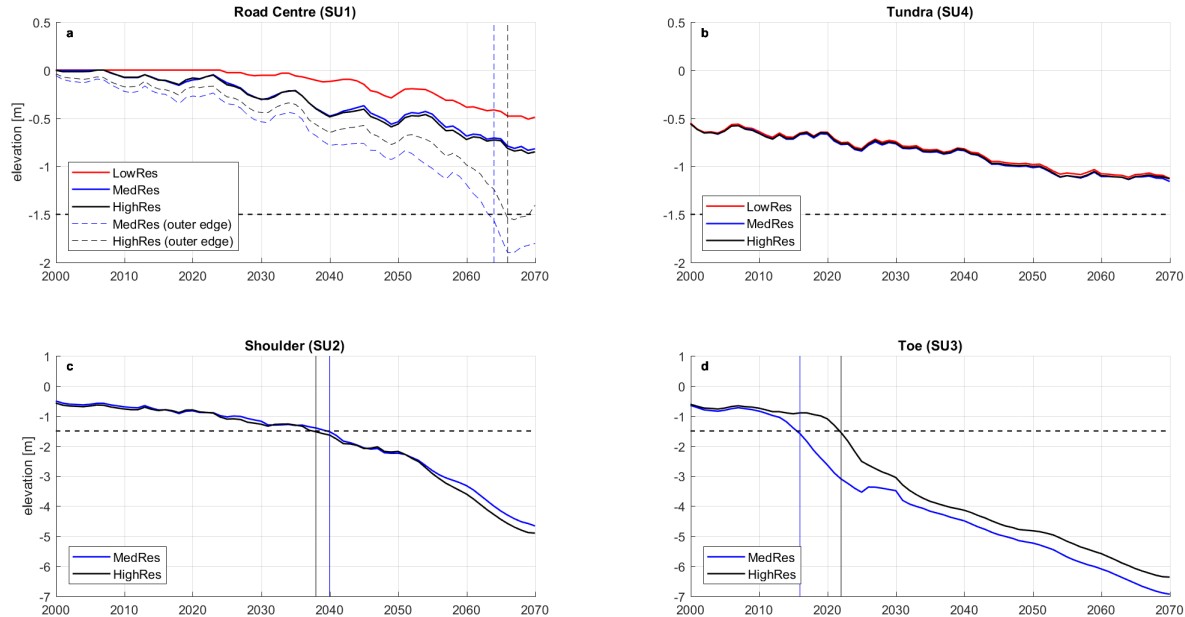

825

**Figure 6: Temporal evolution of the maximum thaw depth (MTD) under gravel road (a,c) and tundra conditions (b,d) for different lateral model resolutions. Shown is the 5 year mean of maximum thaw depth (MTD) under RCP8.5 warming for low horizontal model resolution (2 tiles, red lines), medium resolution (vulnerable setting, 5 tiles, blue curves), and high resolution (vulnerable setting, 30 tiles, black curves). Also shown is the outer edge of the road centre (a, dashed lines). The vertical lines indicate the first occurrence of a MTD exceeding the embankment depth (dashed horizontal line). Note the difference in y-axis scales between upper and lower panels.**

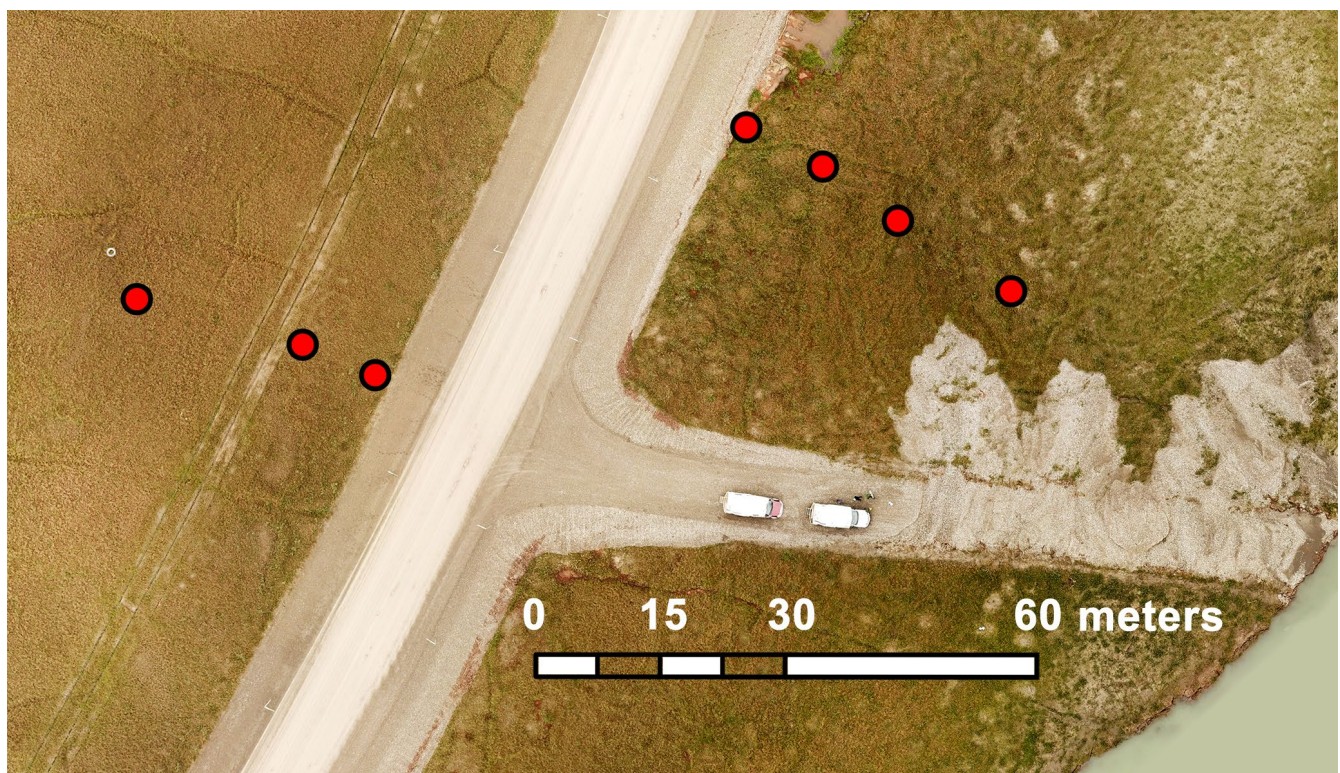

832

Figure A1: Transect of iButton soil temperature loggers across the Dalton highway at our site close to Deadhorse (70.099°N, 148.511°W). Red dots indicate the position of ibutton sensors, the bright grey element shows the gravel road surface, the adjacent darker grey elements show the road shoulders. To the right a side road leading to the adjacent Sagavanirktok River (lower right) is seen (drone photo Soraya Kaiser).

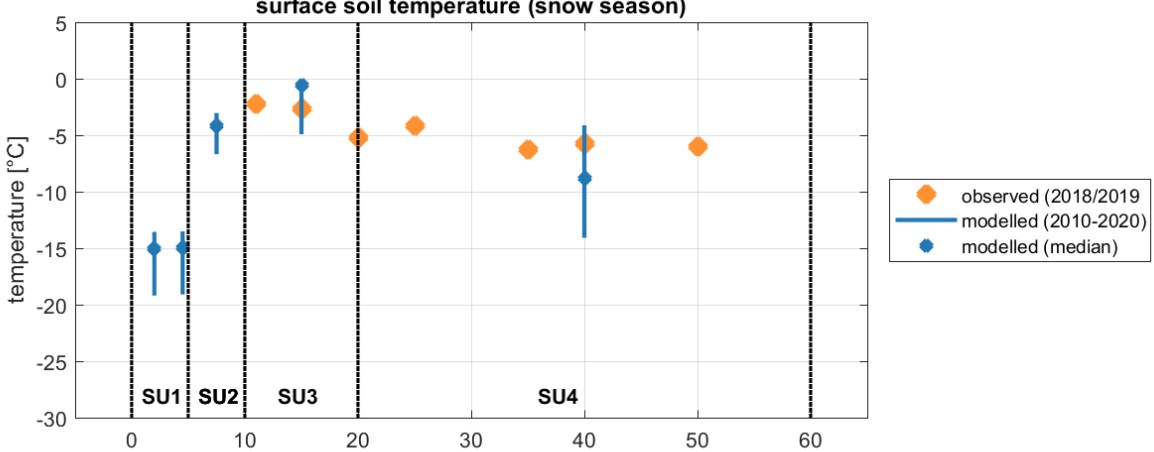

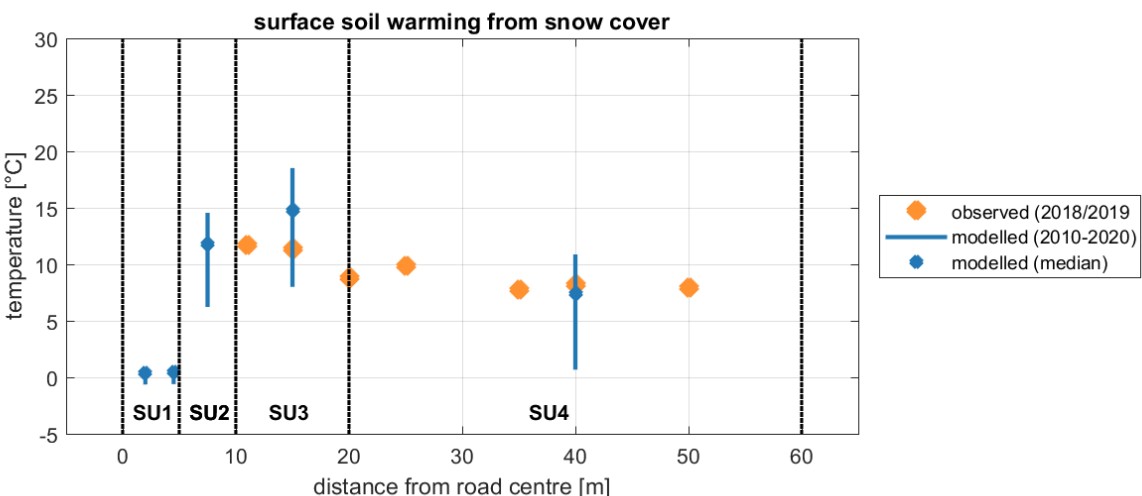


**Figure A2: Transect of modelled (*MedRes* conservative setting) and observed soil surface temperatures for increasing distances to**
**the road centre. The upper panel shows modelled and observed soil surface temperatures averaged over the snow season (1st of**
**November to 30th of April). The lower panel shows the difference of snow-season surface soil minus surface air temperature. Blue**
**dots and blue lines illustrate simulated year-to-year variability and indicate the median and min-max range estimated from 11 snow**
**seasons simulated by CryoGrid3 for the period 2010 to 2020, orange dots give estimates based on ibutton measurements (Dalton**
**highway, Alaska) for the snow season 2018/2019. The vertical dotted black lines illustrate the structural unit domains (SU1: road**
**centre, SU2: embankment shoulder, SU3: toe, SU4: adjacent tundra).**



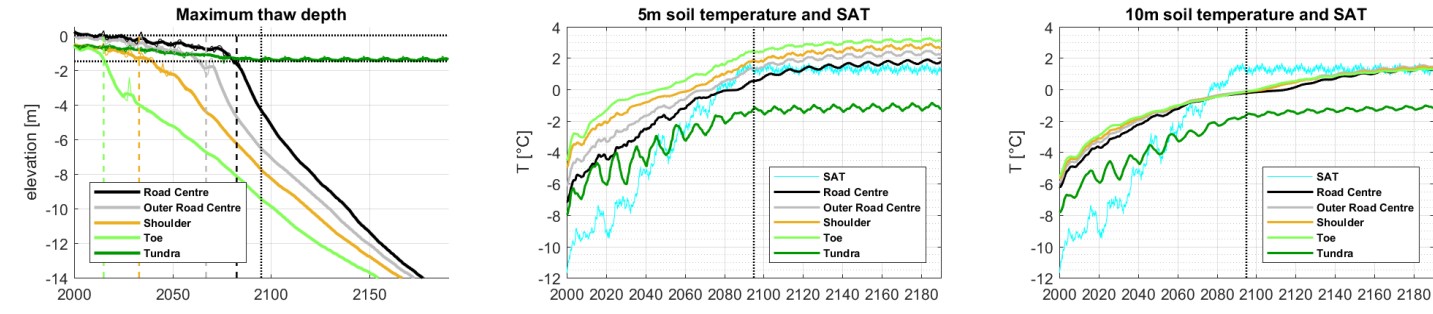


**Figure A3: Climate stabilization under RCP8.5 warming. The left panel illustrates the long-term evolution of maximum thaw depths**
**(MTD) for all structural units for the vulnerable setting (MedRes, 5 tiles, see also Fig.5). The middle and right panels show soil**
**temperatures at 5 meter and 10 meter depth below the tundra surface for the different structural units along with surface air**
**temperature (5 year moving means). Black dotted vertical lines at 2095 correspond to the beginning of climate stabilization.**



| Depth [m] | Initial water/ ice content | Mineral | Organic | Porosity | Type |
|---|---|---|---|---|---|
| **Tundra** | | | | | |
| [0..0.3] | 0.7 | 0.05 | 0.2 | 0.75 | Organic |
| [0.3..2.3] | 0.4 | 0.55 | 0.05 | 0.4 | silt |
| [2.3..10] | 0.4 | 0.55 | 0.05 | 0.4 | Sand |
| >10 | 0.3 | 0.7 | 0.0 | 0.3 | Sand |
| **Road Centre** | | | | | |
| [-2.5..-2.4] | 0.05 | 0.8 | 0.0 | 0.2 | fine grained gravel |
| [-2.4..1.5] | 0.2 | 0.7 | 0.0 | 0.3 | coarse grained gravel |
| [1.5..2.3] | 0.4 | 0.55 | 0.05 | 0.4 | Silt |

| [2.3..10] | 0.4 | 0.55 | 0.05 | 0.4 | Sand |
| --- | --- | --- | --- | --- | --- |
| >10 | 0.3 | 0.7 | 0.0 | 0.3 | Sand |

**Table 1: Ground stratigraphies with volumetric fractions of the ground constituents. Negative depths correspond to layers elevated**
**relative to the tundra surface at 0 m. The setting for the road shoulder is identical to the road centre, besides lacking a fine grained**
**surface layer and starting from elevations between the road surface and the tundra (see Fig. 3). The road embankment base is**
**assumed at 1.5 m depth. In the case of ponding at the toe ('vulnerable' setting), we assume high excess ice in the ground between 1**
**m and 2 m depth.**



| albedo | snow free surface tundra | 0.2 | Langer et al. (2011) |
| --- | --- | --- | --- |
| albedo | pond | 0.07 | Burt (1954) |
| albedo | snow free surface gravel road | 0.3 | (Andersland and Ladanyi, 1994) |
| albedo | fresh snow | 0.85 | (Grenfell and Maykut, 1977) |
| albedo | old snow | 0.5 | (Grenfell and Maykut, 1977) |
| density | snow cover | 250 kg m$^{-3}$ | (Sturm et al., 2010) |
| thermal conductivity | mineral soil and gravel fraction | 3.0 W m$^{-1}$K$^{-1}$ | (Langer et al., 2013),(Farouki et al., 1982) |
| thermal conductivity | organic soil fraction | 0.25 W m$^{-1}$ K$^{-1}$ | (Farouki et al., 1982) |
| hydraulic conductivity | below surface ground | 1e$^{-5}$ m sec$^{-1}$/ | (Boike et al, 2019) |
| volumetric heat capacity | mineral soil and gravel fraction | 2.0*10$^6$ J K$^{-1}$ m$^{-3}$ | Farouki (1982) |
| volumetric heat capacity | organic soil fraction | 2.5*10$^6$ J K$^{-1}$ m$^{-3}$ | Farouki (1982) |

| geothermal heat flux | lower model boundary | 0.05 W m$^{-2}$ | (Lachenbruch et al., 1982) |
|---|---|---|---|

Table 2: Model parameters in CryoGrid3 used in this study. Effective thermal conductivities and heat capacities of each model layer are calculated based on the volumetric fractions of the ground constituents water, ice, air, mineral, gravel, and organic (Cosenza et al.,2003;Westermann et al., 2013). The temperature dependence of the effective thermal conductivity and capacity is taken into account by calculating temperature-dependent water and ice contents.

| Experiment | Description |
|---|---|
| LowRes | Low resolution run with 2 structural units (2 tiles) |
| | Tile 1: road centre (10 m width), Tile 2: tundra (90 m width) |
| | Toe and shoulder are not resolved (i.e. no snow accumulation, no ponding, no increased incident solar radiation) |
| MedRes | Medium resolution run with 4 structural units (5 tiles) |
| | Tile 1: road centre (4 m width), Tile2: outer road centre (1 m width), Tile3: shoulder (5 m width), Tile 4: toe (10 m width), Tile 5: tundra (40 m width) |
| 'conservative' | Snow accumulation at shoulder and toe are represented |
| 'vulnerable' | Same as 'conservative setting, but additional ponding at toe and increased incident solar radiation at shoulder (southern facing) |
| HighRes | High resolution run with 4 structural units (30 tiles) |
| | Tile 1-5: road centre (1 m width),Tile 6-10: shoulder (1 m width), Tile 11-20: toe (1 m width), Tile 21-25: tundra (1 m width), Tile 26-27: tundra (5 m width), Tile 28-29: tundra (10 m width), Tile 30: tundra (45 m width) |
| 'conservative' | Not performed for HighRes |
| 'vulnerable' | Snow accumulation at shoulder and toe (snow height depending on distance to road), ponding at toe, increased incident solar radiation at shoulder (southern facing) |
| Ref | Tundra reference run without infrastructure (using the same tundra stratigraphy as in the other experiments) |

Table 3: Description of performed simulation experiments with CryoGrid3