# Peer review of "Consequences of permafrost degradation for Arctic"

_The Cryosphere, 2020_

## Referee Comment (RC1) · Anonymous Referee #1 · 27 Oct 2020

I enjoyed reading this paper, in which the authors propose the use of process-based tiling models (PTMs) to study the decade-scale impact of infrastructure placement on permafrost stability. Through a case study at the Dalton Highway in northern Alaska, the authors demonstrate the feasibility of PTMs for bridging the spatial gap between computationally expensive, short timescale geotechnical models (GTMs) and coarse-resolution land surface models (LSMs). The authors apply process-based tiles within Cryogrid-3, an approach which has been used previously to model permafrost dynamics in other environments where lateral fluxes of water and energy are important.

[Figure]

However, the present application of PTMs to infrastructure is novel, and the authors propose that it could be used to conduct pan-Arctic risk assessments for threats to infrastructure under different climate change scenarios. In general, I thought that the paper is well written, and the authors present an effective case study of the PTM approach at their field site. However, two general concerns I have are that: 1) I would like the authors to elaborate more on the outline they propose in section 4 for how to expand simulations like those reported here to a pan-Arctic risk assessment, and 2) I would appreciate a bit more discussion of the permafrost physics in the simulations from their Dalton Highway case study. In addition, I have several more specific comments, listed below.

General comments:

1) I appreciate that the authors sketch out how to incorporate PTMs into pan-Arctic risk assessments in section 4, but currently the proposed workflow is somewhat difficult to understand. I know that this section is speculative, but I'd like a bit more detail about the connection between LSMs and PTMs. The authors state that results from an initial set of PTMs "could ... be used by LSMs to define a tiling-based model-setup for describing infrastructure in a reduced but manageable manner." Does this mean that a pan-Arctic LSM simulation would be executed, incorporating a sub-grid tiling scheme that accounts for the location of infrastructure? And if so, how would this be more informative than simply using meteorological forcing data derived from ESMs in the initial set of PTMs?

2) I am also curious at what stage of the analysis it is most appropriate to assess uncertainty related to the site-specific manners in which infrastructure may accelerate permafrost thaw. The authors acknowledge that "a specific gravel road can deviate strongly from our assumed setting here." As they state, one source of variability is road geometry, and another source of variability is that gravel roads can cause thaw through a variety of mechanisms, such as low-albedo dust deposition on the tundra or impedance to surface drainage. Another important factor to consider is that, unlike

the case study presented here, the effects of these process are often asymmetrical, occurring preferentially on one side of the roadway (e.g., Raynolds et al., 2014; Abolt et al., 2017). It seems like uncertainty in each of these factors would have a big impact on the risk assessment. Is it best to account for it within PTMs, LSMs, or GTMs?

3) The authors state in the abstract that rates of permafrost thaw adjacent to the high-way follow "a two-phase beahvior", and in section 3.2, they state that thaw rates sharply increase "once a critical warming of the ground has been reached." This is an important result and I think it would be worthwhile for the authors to elaborate more on when this threshold occurs. For example, does the abrupt increase in thaw coincide with events such as the formation of an open talik, or the onset of surface water ponding—or does it lag several years behind? Also, does the system stabilize by the end of the simula-tions? This information would be informative to the risk assessment, and I also think it would be interesting in the context of other numerical simulations of thermokarst.

Specific comments:

1) 72-74: This sentence is vague. I assume the issues referred to are related to infras-tructure stability, but please be more specific.

2) 149-151: Please re-write this sentence or break it into two shorter ones.

3) 167-168: Please be more specific. Are you referring to improvements in subsurface physics that would allow ice lens formation to be simulated?

4) 176-178: This sentence is difficult to understand. Please consider rewriting it.

5) 183-185: Please include a concise definition of Process-based tiling models toward the start of this paragraph, including how they differ from the virtual tiling methods mentioned in the previous paragraph.

6) 199: Replace the phrase "pan-Arctic wide" with "at the pan-Arctic scale."

7) 205: Replace the phrase "we chose this region as our target region for modeling"

[Figure]

with something like "we chose this region for our case study."

8) 221: Even though it's obvious, please begin this paragraph with a simple topic sentence, such as "Although our model was designed to capture essential components of permafrost thaw, it is limited by several physical assumptions." Also, you might consider specifying either here or in section 2.2 that your simulations are set up such that the primary driver of thaw is enhanced snow accumulation adjacent to the highway, as opposed to dust deposition or the backing up of surface flow.

9) 222-224: Please be more specific about how the constant bulk density of snow affects the subsurface. I assume this means that thermal conductivity is constant as well?

10) 236: Please define the acronym SNAP, and specify the time range for the forcing data here.

11) 264-265: I suggest rewording this to say "results in markedly warmer winter temperatures."

12) 366: Consider deleting the phrase "once a critical level of ground warming has been reached," as the first half of the sentence already references the idea of a threshold. In this paragraph, consider elaborating on when this threshold is reached.

13) What is an "external water flux"? Is this analogous to precipitation? Or lateral flow? And does it occur throughout the spatial domain? How did you settle on the amount of 2 mm per day?

Papers cited:

Raynolds et al., 2014. Cumulative geoecological effects of 62 years of infrastructure and climate change in ice-rich permafrost landscapes, Prudhoe Bay Oilfield, Alaska. Global Change Biology, 20, 1211-1224.

Abolt et al., 2017. Numerical modeling of ice wedge polygon geomorphic transition.

Permafrost and Periglacial Processes, 28, 347-355.

---

## Referee Comment (RC2) · Anonymous Referee #2 · 9 Dec 2020

General comments.

The context of this study is to analyze consequences of permafrost degradation on Arctic infrastructure using a model and an approach which attempts to bridge the gap between small and large scales and processes. This is useful because most global scale/land surface models cannot resolve the detail typically required for small-scale systems, such as the case of infrastructure impacts considered here. The study suggests a Process-based Tiling Model approach to bridge this gap, which attempts to capture sufficient detail but which also can be scaled up to larger LSM-scale with rela-

tive ease. The study uses measurements from a road site in northern Alaska, and involves model-based analysis of the impact the road has on permafrost thaw undergoing climate warming. The study combines methodological development with site-specific analysis in the form of a case study.

This study is very relevant and should be of value for readers of TC, especially those interested in improving representation of LSMs for arctic processes. My main concern is that the PTM approach/Cryogrid is only evaluated against itself. If one of the objectives of the study is to present methodology development of the PTM approach, then a more robust evaluation and validation would be useful. Currently, different Cryogrid models are configured which are meant to correspond to simpler versus more complex conceptualization of processes, as well as finer versus coarser spatial resolution of the road-bank-tundra system. This is certainly of value and such comparisons are useful. However, since the evaluations are only conducted within the framework of the Cryogrid model, there is no external validation of the approach against other/independent models, for example models which account for higher resolutions and additional physical processes relevant at the small (100-m) scale.

Hence, a validation of the approach in the strict sense is currently lacking. This makes it difficult to place the actual performance, robustness and accuracy of the Cryogrid/PTM approach in a broader context. An evaluation/validation against a geotechnical model/GTM, or other similar fine-scale process model, would make the contribution stronger. This is important, because some process relevant at the small scale are not accounted for. These generally include consideration of later heat flux, both advective and conductive (2D), and possibly subsurface lateral flow of water (Darcy's law is mentioned, but not clear if flow can occur both in the vertical and transverse direction, as 2D flow), as well as mechanical soil deformations potentially leading to ground subsidence, which typically occur during freeze-thaw cycles. Such processes impact heat distribution in the ground. Also, only a 2D transect of the subsurface is considered, but flow and heat flux in the transverse direction (3D model) could also be

relevant. Of course the ambition of the PTM approach is to avoid accounting for detailed processes/representations. Therefore, it would benefit readers to know precisely how accurate the PTM approach actually is, by comparison against a model or models which account for them. This would then more robustly support the PTM approach in being sufficiently adequate for "bridging the gap" to LSM scales.

Also, some of the details of the implementation of the model could be clarified. For example, the ponding feature in the model needs clarification, please see comments below. This is especially important considering its impact on results, for example Fig 5. Also, the implementation of effective thermal conductivity could be clarified, specifically if/how the dynamic phase state of water, as ice or liquid, filling the pore space is accounted for in the subsurface, or if only a static thermal conductivity of the different materials (shown in Table 2) is considered.

Further, the presentation would benefit from an explanation of what the advantage of the approach is compared to using, for example, a GTM at the small scale which is forced/driven by boundary conditions obtained from an LSM? This is essentially what is being done in this study. Hence, infrastructure impacts could, at least in principle, be modeled that way by combining large and small scales using different models, and hence without bypassing fine-scale processes and their effects. Perhaps this is the intention of the discussion in Section 4; if so, that section could use clarification, please see comment below.

Specific comments.

L 106-107: There are other lateral processes which also impact permafrost change and thaw, especially subsurface lateral processes such as flow of water and associated advected heat flux, both for 2D and 3D model representations.

L 110: It is not entirely clear what a "conservative assessment" means in the context of "permafrost thaw impacts". If models are conservative in their assessments of the impacts of permafrost thaw, then one would assume they are over-estimating thaw

Interactive
comment

rates. Perhaps this part could be rephrased/clarified.

L 129-130: There are several other studies on modelling polygonal tundra dynamics, please consider citing.

L 180-190: Are there any other PTM models other than Cryogrid, if so please consider citing.

L 245: Vulnerable case, ponding – please explain/clarify exactly how ponding is represented in the model and what effects it has. Does this impact the SEB? Enable infiltration of water from the surface, as a source? Does it impact thermal properties, is effective thermal conductivity considered at the surface-subsurface interface? etc.

Section 2. Model setup, boundary conditions. Would be useful to mention here that the model depth is 1000 m (if this indeed is the case), this is "hidden away" in Appendix A3. The reason being a thin model domain of only approx 10 m (as Fig 3 and 4 depict) would most likely incur boundary effects from the base boundary condition.

Section 3/Figures 4,5: It would also be nice to see/visualize the boundary conditions which are used to drive the model, if possible, for example the forcing/driving conditions at the surface. This can help understand and interpret the results shown in these figures.

Section 4. The model and approach used in this paper is very nice and clearly relevant, but the purpose of this section is not so clear. It would be far more informative and convincing to actually demonstrate the strategy for the different applications suggested here. For example, can a demonstration of a model linking to LSMs beyond the 100-m scale modeled here be made? Also, the discussion on suggested analysis for risk assessment from line 350 onwards would have more substance if a demonstration could be made.

Fig 4: Consider making a note that the vertical extent (elevation) of the model domain is not depicted here, could refer to Appendix A3. Also in Fig 3, the extent of the model

[Figure]

in the vertical direction is depicted only as around 10 m, but according to A3 is 1000 m.

Fig 6: Please improve clarity, font sizes and legends are relatively small.

Table 2: Hydraulic properties are not described; what is the hydraulic conductivity or permeability, and porosity, of the different materials used? Typically, there is great variability and hence uncertainty in hydraulic conductivity and this greatly impacts water flow.

Table 2: Do the thermal conductivities refer to dry porous media? How is effective thermal conductivity, accounting for phase state of water-ice in the porous media, calculated?

Table 3: Is there a 'conservative' setting for the HighRes model setup? Please clarify.

Fig A2: What do the model ranges, min to max, with median, refer to here? What is changed between the "Cryogrid model runs" to get the ranges of results? Please clarify.

---

## Author Comment (AC1) · 25 Feb 2021

We thank the reviewers for their constructive comments which are very helpful for improving our manuscript. Based on the comments made by both reviewers, we have especially reworked our perspective section. We also extended our model simulations to explore long-term model behaviour beyond the year 2100 and investigated further model diagnostics to explore in more detail the model physics discussed, especially with respect to talik formation. In the following we reply in detail to all the comments made:

Reviewer 1: ...However, two general concerns I have are that: 1) I would like the authors to elaborate more on the outline they propose in section 4 for how to expand simulations like those reported here to a pan-Arctic risk assessment, and 2) I would appreciate a bit more discussion of the permafrost physics in the simulations from their Dalton Highway case study. In addition, I have several more specific comments, listed below.

General comments: 1) I appreciate that the authors sketch out how to incorporate PTMs into pan-Arctic risk assessments in section 4, but currently the proposed work-flow is somewhat difficult to understand. I know that this section is speculative, but I'd like a bit more detail about the connection between LSMs and PTMs. The authors state that results from an initial set of PTMs "could ... be used by LSMs to define a tiling-based model-setup for describing infrastructure in a reduced but manageable manner." Dtalikoes this mean that a pan-Arctic LSM simulation would be executed, incorporating a sub-grid tiling scheme that accounts for the location of infrastructure? And if so, how would this be more informative than simply using meteorological forcing data derived from ESMs in the initial set of PTMs?

We have intensively reworked section 4 to substantiate our suggested strategy for connecting PTMs with LSMs and GTMs (see below, end of general comments reply) where we now stress more clearly the difference in applicability and informative values between LSM and GTM analyses. We now sketch out three main lines of connecting PTMs and LSMs/ESMs

1) PTMs can support LSM development for defining suitable tiling-based descriptions of infrastructure such that LSMs can be used for calculating large-scale estimates of potential risks regarding infrastructure failure. LSMs would not account for the exact location of infrastructure, but rather for the general presence of infrastructure within a LSM grid cell. Infrastructure in each grid cell is then subject to the climatic conditions taken from the corresponding atmospheric fields of each LSM grid cell (rather than to site-specific local climate). 2) PTMs could be used as an offline add-on in LSMs for

capturing infrastructure. 3) PTMs could be run in stand-alone mode driven by climate conditions from an ESM (like done in this study). This allows the accounting of more site-specific aspects (e.g. local soil stratigraphy, bias-corrected climate forcing, etc.) For details see new section 4 below.

2) I am also curious at what stage of the analysis it is most appropriate to assess uncertainty related to the site-specific manners in which infrastructure may accelerate permafrost thaw. The authors acknowledge that "a specific gravel road can deviate strongly from our assumed setting here." As they state, one source of variability is road geometry, and another source of variability is that gravel roads can cause thaw through a variety of mechanisms, such as low-albedo dust deposition on the tundra or impedance to surface drainage. Another important factor to consider is that, unlike the case study presented here, the effects of these process are often asymmetrical, occurring preferentially on one side of the roadway (e.g., Raynolds et al., 2014; Abolt et al., 2017). It seems like uncertainty in each of these factors would have a big impact on the risk assessment. Is it best to account for it within PTMs, LSMs, or GTMs?

We agree that these factors are all critical for judging infrastructure failure at a given location. For such cases, the capturing of these processes likely requires the use of complex fully coupled thermal-hydrological 3D modelling. If the focus is more on a regional or pan-arctic risk assessment, site-specific factors are of less importance but it will be crucial to describe controlling factors (such as ponding next to a road) which determine overall infrastructure failure on a regional scale. The potential risk will not depend on where exactly infrastructure will fail, but on the occurrence of infrastructure failure under climatic conditions typical for a given grid cell. Such analyses could be performed with PTM-LSM setups.

In the revised section 4 we also sketch out in more detail how PTMs can support GTM analyses for determining infrastructure failure under climate change.

3) The authors state in the abstract that rates of permafrost thaw adjacent to the highway follow "a two-phase behavior", and in section 3.2, they state that thaw rates sharply increase "once a critical warming of the ground has been reached." This is an important result and I think it would be worthwhile for the authors to elaborate more on when this threshold occurs. For example, does the abrupt increase in thaw coincide with events such as the formation of an open talik, or the onset of surface water pondingâËŸA ËĞ-Tor does it lag several years behind? Also, does the system stabilize by the end of the simulations? This information would be informative to the risk assessment, and I also think it would be interesting in the context of other numerical simulations of thermokarst.

We consider these aspects interesting diagnostics for the model result interpretation and have analyzed talik formation dynamics and performed a climate stabilization scenario to investigate long-term thaw dynamics. Our analyses show that the timing of increase in thawing rates can occur well before full talik formation. Yet the choice of a less strong criterion for soil refreeze allows us to broadly track the two phase behaviour. We achieve this by determining the timing of the formation of a partially unfrozen soil layer, where refreeze does not exceed 25% of pore water (see the timing indicated by dashed vertical lines in the updated Figure 5 below). In the manuscript we now discuss this aspect as follows: "... Thawing rates can be split into two periods: a period of slow and gradual increase in maximum thaw depths, followed by a sharp increase in thawing rates once a critical warming of the ground has been reached. The two-phase thawing characteristic cannot be explained through changes in external climate forcing (see Figure A3), but by internal dynamics following the formation of year-round partially unfrozen layers in the ground. We can roughly track the timing of the increase in thawing rates by the first occurence of a layer with a maximum refreeze of 25% of the pore water (dashed vertical lines, Figure 5). This diagnostic can be seen as a precursor for later full talik formation (i.e. year-round unfrozen conditions).

We now also discuss in section 3.2 in more detail the physical reasons for the difference in pace of thaw between the different structural units: "The outer road edge is destabilized around the year 2060 (2075 without ponding). Talik formation under the

road center only occurs after year 2080, but triggers a very pronounced increase in thawing rates (Fig. 5). The large increase in late-century thawing rates under the road center are a consequence of previous continuous warming of subgrade temperatures through lateral heat flux in depth. This heat flux leads to an almost isothermal temperature depth profile under the road close to 0°C (see Fig.4, lowest panels), making the ground strongly vulnerable to further warming."

For analysing long-term consequences of climate stabilization, we have extended our model-setup and run our model under the assumption that climate will stay stationary at the end of 21st century RCP8.5 climate conditions. In Figure A3 we discuss the evolution of deep soil temperature (at 5m and at 10m depth) together with surface air warming. We now show that tiles affected by infrastructure show continuously deepening thaw depths beyond 2100, while the tundra tile shows stabilization. In the manuscript we discuss this aspect accordingly:

[revised manuscript text omitted]

Specific comments: 1) 72-74: This sentence is vague. I assume the issues referred to are related to infrastructure stability, but please be more specific.

Yes, the focus is on engineered structures, considering risks of general permafrost degradation and thaw settlement. We will specify this information in the revised manuscript.

2) 149-151: Please re-write this sentence or break it into two shorter ones.

We now broke this sentence into two shorter sentences.

3) 167-168: Please be more specific. Are you referring to improvements in subsurface physics that would allow ice lens formation to be simulated?

We refer to interactive ice lens growth as part of a beneficial future model development idea, however, the current model capacity does not allow ice to grow as ice lenses.

4) 176-178: This sentence is difficult to understand. Please consider rewriting it.

We now state: "The benefit of these developments will be implemented such that the models can help understand and quantify permafrost thaw-associated feedbacks to the climate system. Such feedbacks can only be systematically quantified using the coupled framework of Earth System Modeling, which no other models discussed in this study can represent."

5) 183-185: Please include a concise definition of Process-based tiling models toward the start of this paragraph, including how they differ from the virtual tiling methods mentioned in the previous paragraph.

As opposed to "virtual tiling" approaches as mentioned in the previous LSM section, we here refer to "process-based tiling" by considering tiling concepts which account for the dynamic lateral interaction between individual tiles. For avoiding confusion with different uses of terms, we now avoid the use of "virtual" and additionally shortly discuss the use of "adaptive tiling". We have modified the text accordingly:

Section 1.2.2.: "An established LSM strategy for representing landscape heterogeneity (e.g. for. representing excess ice) consists of splitting grid cells into "tiles" to take into account sub-grid variability in surface characteristics, but such tiling schemes are not spatially explicit and do not describe interaction between tiles" Section 1.2.3: ..."As opposed to standard LSM "tiling" approaches as mentioned in the previous section, we here refer to "process-based tiling" by considering the dynamic lateral interaction between individual tiles. Tiling approaches can be further optimized in efficiency by applying "adaptive tiling" concepts with using a dynamical number of tiles (Fisher, Koven 2020).

6) 199: Replace the phrase "pan-Arctic wide" with "at the pan-Arctic scale." We will do so.

7) 205: Replace the phrase "we chose this region as our target region for modeling" with something like "we chose this region for our case study." We will do so.

8) 221: Even though it's obvious, please begin this paragraph with a simple topic sentence, such as "Although our model was designed to capture essential components of permafrost thaw, it is limited by several physical assumptions." Also, you might consider specifying either here or in section 2.2 that your simulations are set up such that the primary driver of thaw is enhanced snow accumulation adjacent to the highway, as opposed to dust deposition or the backing up of surface flow. We will include these

suggestions.

9) 222-224: Please be more specific about how the constant bulk density of snow affects the subsurface. I assume this means that thermal conductivity is constant as Well? Thermal conductivity of snow depends on the fractions of air, water, and ice in a given snow layer in Cryogrid. Infiltration and refreezing of rain or meltwater changes the water to ice ratio and accordingly affects the effective thermal conductivity and capacity of the snow layer. We now added this information in the text: ..."Our model represents the thermal impact of the snow cover on the ground by a simple bulk snow density approach neglecting any temporal changes in snow cover properties resulting from snow metamorphism (i.e., depth hoar) or wind compaction (i.e., ice crusts). However, the effective thermal properties of the snow layer can change following infiltration and refreezing of rain or meltwater."

10) 236: Please define the acronym SNAP, and specify the time range for the forcing data here. We have specified the forcing data reference: "Down-scaled climate data (SNAP (SCENARIOS NETWORK FOR ALASKA and ARCTIC PLANNING, (Lader et al., 2017) was used to force our model under historical, present day and projected climate change, covering the period from 1975 to 2100."

11) 264-265: I suggest rewording this to say "results in markedly warmer winter temperatures." We will use this rewording.

12) 366: Consider deleting the phrase "once a critical level of ground warming has been reached," as the first half of the sentence already references the idea of a threshold. In this paragraph, consider elaborating on when this threshold is reached. We agree on deleting the last part of the phrase. We now have analysed occurrences of talik formation for explaining the threshold behaviour (see comments above).

13) What is an "external water flux"? Is this analogous to precipitation? Or lateral flow? And does it occur throughout the spatial domain? How did you settle on the amount of 2 mm per day? By "external water flux" we describe a water flux from an

assumed adjacent reservoir which could e.g. represent a surficial water flux. This flux is only applied to the tundra and toe tiles (i.e. not to infrastructure tiles elevated above the natural tundra surface). We inferred a minimum flux value of 2mm/day to achieve closed to saturated conditions in the tundra soil. Without this additional flux, our simulated soil moisture content (which was solely determined by snowmelt, rain, and evaporation) was too dry. We now added information about our model setting accordingly: ..."We prescribe an external water flux of 2 mm day-1 for the tundra and toe tiles for the period of unfrozen soil surface conditions. This flux could mimic the impact of surficial lateral water fluxes or could be understood as a correction factor for precipitation biases. We introduce this flux for capturing observed high soil moisture conditions in tundra soils next to the road at our chosen location."

[Figure]

[Figure]

**Fig. 1.** Temporal evolution of maximum thaw depth (MTD)...Dashed vertical lines indicate the timing of the first occurence of a year-round partially unfrozen layer (maximum refreeze up to 25% of pore water). T

[Figure]

**Fig. 2.** Climate stabilization under RCP8.5 warming. The left panel illustrates the long-term evolution of maximum thaw depths (MTD) for all structural units for the vulnerable setting (MedRes, 5 tiles)....

---

## Author Comment (AC2) · 25 Feb 2021

We thank the reviewers for their constructive comments which are very helpful for improving our manuscript. Based on the comments made by both reviewers, we have especially reworked our perspective section. We also extended our model simulations to explore long-term model behaviour beyond the year 2100 and investigated further model diagnostics to explore in more detail the model physics discussed, especially with respect to talik formation (see below). In the following we reply in detail to all the comments made:

**Referee #2**

General comments The context of this study is to analyze consequences of permafrost degradation on Arctic infrastructure using a model and an approach which attempts to bridge the gap between small and large scales and processes. This is useful because most global scale/land surface models cannot resolve the detail typically required for small-scale systems, such as the case of infrastructure impacts considered here. The study suggests a Process-based Tiling Model approach to bridge this gap, which attempts to capture sufficient detail but which also can be scaled up to larger LSM-scale with relative ease. The study uses measurements from a road site in northern Alaska. and involves model-based analysis of the impact the road has on permafrost thaw undergoing climate warming. The study combines methodological development with site-specific analysis in the form of a case study. This study is very relevant and should be of value for readers of TC, especially those interested in improving representation of LSMs for arctic processes. My main concern is that the PTM approach/Cryogrid is only evaluated against itself. If one of the objectives of the study is to present methodology development of the PTM approach, then a more robust evaluation and validation would be useful. Currently, different Cryogrid models are configured which are meant to correspond to simpler versus more complex conceptualization of processes, as well as finer versus coarser spatial resolution of the road-bank-tundra system. This is certainly of value and such comparisons are useful. However, since the evaluations are only conducted within the framework of the Cryogrid model, there is no external validation of the approach against other/independent models, for example models which account for higher resolutions and additional physical processes relevant at the small (100-m) scale. Hence, a validation of the approach in the strict sense is currently lacking. This makes it difficult to place the actual performance, robustness and accuracy of the Cryogrid/ PTM approach in a broader context. An evaluation/validation against a geotechnical model/GTM, or other similar fine-scale process model, would make the contribution stronger. This is important, because some process relevant at the small scale are not accounted for. These generally include consideration of later
heat flux, both advective and conductive (2D), and possibly subsurface lateral flow of water (Darcy's law is mentioned, but not clear if flow can occur both in the vertical and transverse direction, as 2D flow), as well as mechanical soil deformations potentially leading to ground subsidence, which typically occur during freeze-thaw cycles. Such processes impact heat distribution in the ground. Also, only a 2D transect of the subsurface is considered, but flow and heat flux in the transverse direction (3D model) could also be relevant. Of course the ambition of the PTM approach is to avoid accounting for detailed processes/representations. Therefore, it would benefit readers to know precisely how accurate the PTM approach actually is, by comparison against a model or models which account for them. This would then more robustly support the PTM approach in being sufficiently adequate for "bridging the gap" to LSM scales.

We see the point that we do not provide a model validation in a "strict sense". The general problem we are facing with such a validation are different model foci of GTMs (small-scale, spatial detail) and PTMs (process-oriented): GTMs often focus on describing construction specific aspects in high detail, but miss out other factors such as snow pack dynamics. PTMs on the other hand miss construction details but describe impacts from changing boundary conditions (such as a variable snow pack). Therefore, a direct comparison of results inferred from these model classes is difficult and has to be interpreted with caution. Using fully coupled hydro-thermo-mechanical GTMs requires complex parameter fitting to experimental data for appropriate model setups. Given that these models are still novel and lack experience values, large parameter uncertainties remain which further impede model comparisons. In our study we qualitatively compare our results e.g. with GTM results from Fortier et al. 2011 and Flynn et al. 2016. Both studies investigate permafrost degradation for a gravel road site on discontinuous permafrost. Despite a roughly comparable model-setup, these studies analyse a system starting from much warmer soil conditions compared to our considered continuous permafrost site at Deadhorse. Furthermore, assumptions with e.g. regard to snow accumulation and ponding differ among our studies - factors which strongly affect quantitative estimates. Differences in model-setups do

**TCD**
therefore not allow for a quantitative comparison. Even under identical boundary conditions, we expect that differences between our model results and fine-scale models can be large, if site-specific processes such as advective heat transport from lateral water flow are considered only in one model setup. We therefore further stress in the revised manuscript the need of using fine-scale models for a site-specific risk assessments (we do not recommend to do such an assessment with using a PTM, see revised section 4 below, end of general comments reply).

In the revised manuscript, we now also have included more aspects for a qualitative check of our results and cite additional GTM studies. Further, we now discuss in more detail the physical processes which determine the degradation dynamics (see comments to reviewer 1). We also added additional information in the model description section to underline which processes are captured in our model, and which processes are missing. In short reply to the points raised above, this concerns: Lateral heat flux and subsurface lateral flow of water

Our version of Gryogrid3 describes conductive vertical and lateral heat flux between tiles and describes vertical and lateral water flux. Yet our model does not describe the effect of heat advection from water flow which is subject for further model development. We do not consider the omission of modelling advective heat transport critical for our considered case here, as we focus our discussion on the timing when ground thaw reaches the subgrade underneath the embankment for the first time. Advective heat transport will increasingly become important at later stages of permafrost degradation when pathways of subsurface flow will develop and contribute to higher rates of degradation. If our focus was on modelling a gravel road on warm, discontinuous permafrost, advective heat transports effects are likely to play a much more important role for near-term permafrost degradation.

3D flow and heat flux

We have constrained our model setup to an idealized 2D linear infrastructure case
to capture the dominant road-permafrost interactions stemming from snow-free conditions at the road center, snow accumulation on the shoulders, and ponding at the toe. In reality, the thermal regime of a road can be affected by factors requiring a 3D model setup. In our revised manuscript we underline the need for GTM modelling to capture such effects (see new section 4), but an explicit modelling of 3D features with Cryogrid would go beyond the applicability the model is designed for.

Also, some of the details of the implementation of the model could be clarified. For example, the ponding feature in the model needs clarification, please see comments below. This is especially important considering its impact on results, for example Fig 5. Also, the implementation of effective thermal conductivity could be clarified, specifically if/how the dynamic phase state of water, as ice or liquid, filling the pore space is accounted for in the subsurface, or if only a static thermal conductivity of the different materials (shown in Table 2) is considered.

We now give more details for the implementation of ponding in the model set-up section (2.2): ... "Pond formation has a pronounced effect on our modeled soil temperatures by altering surface energy fluxes through lowering surface albedo and replacing thermal properties of the soil surface by those of a water body." We now also specify the albedo value of a water surface (0.07) in table 2.

Effective thermal conductivity is calculated in the model at each time step depending on the soil constituents in each layer - i.e. the fractions of mineral/gravel and organic, water and/or ice fraction, and air. We now added the information about calculation of effective thermal conductivity and capacity in the legend of table 2: ...Effective thermal conductivities and heat capacities of each layer are calculated based on the volumetric fractions of the ground constituents water, ice, air, mineral, gravel, and organic (Cosenza et al.,2003;Westermann et al., 2013).

Latent heat effects through phase change are explicitly considered when modelling heat diffusion with Cryogrid.

TCD
Further, the presentation would benefit from an explanation of what the advantage of the approach is compared to using, for example, a GTM at the small scale which is forced/driven by boundary conditions obtained from an LSM? This is essentially what is being done in this study. Hence, infrastructure impacts could, at least in principle, be modeled that way by combining large and small scales using different models, and hence without bypassing fine-scale processes and their effects. Perhaps this is the intention of the discussion in Section 4; if so, that section could use clarification, please see comment below.

Based on the comments made by both reviewers, we have strongly revised section 4 for substantiating our proposed strategy of combining PTMs with LSMs and GTMs.

Revised section 4:

[revised manuscript text omitted]

Specific comments. L 106-107: There are other lateral processes which also impact permafrost change and thaw, especially subsurface lateral processes such as flow of water and associated advected heat flux, both for 2D and 3D model representations.

We wanted to give a few examples of lateral processes and now added the aspect of subsurface water flow.
L 110: It is not entirely clear what a "conservative assessment" means in the context of "permafrost thaw impacts". If models are conservative in their assessments of the impacts of permafrost thaw, then one would assume they are over-estimating thaw rates. Perhaps this part could be rephrased/clarified.

We now rephrased this sentence: Thus, current model assessments are most likely underestimating permafrost thaw impacts,...

L 129-130: There are several other studies on modelling polygonal tundra dynamics, please consider citing.

We now also cite Pau et al. (2014), Abolt et al. (2018) and Kumar et al. (2016)

L 180-190: Are there any other PTM models other than Cryogrid, if so please consider Citing.

We now also mention work by Jan et al. (2017) in which the authors used a mixed dimension model structure for efficiently simulating surface/subsurface thermal hydrology in low-relief permafrost regions at watershed scales, as well as work by Daanen et al. (2011), in which the authors used the GIPL model for linking large scale climate and permafrost simulations to small scale engineering aspects with a focus on Greenland. We are not aware of further permafrost models being of comparable flexibility and model complexity to Cryogrid.

L 245: Vulnerable case, ponding – please explain/clarify exactly how ponding is represented in the model and what effects it has. Does this impact the SEB? Enable infiltration of water from the surface, as a source? Does it impact thermal properties, is effective thermal conductivity considered at the surface-subsurface interface? Etc.

Yes, pond formation changes the SEB in our model pronouncedly. We now clarified this aspect by adding the following sentence after L245: "...Pond formation has a pronounced effect on our modeled soil temperatures by altering surface energy fluxes through lowering surface albedo and replacing thermal properties of the soil surface by TCD
those of a water body."

We extended table 2 to include the information about our assumed albedo of a pond (0.07). In the model result discussion (section 3.2) we now also mention the effect of ponding on heat uptake: ...." Further, a pond allows for a more efficient heat uptake through vertical mixing within the water column as compared to top-down heat diffusion in solid ground." In the model description section, we now added information about modelling ground subsidence and ponding (with a reference to the work in which the ponding/subsidence scheme is discussed in detail): ...The model explicitly simulates ground subsidence and subsequent ponding as a consequence of melting of excess ice in the ground (Westermann et al., 2016).

Section 2. Model setup, boundary conditions. Would be useful to mention here that the model depth is 1000 m (if this indeed is the case), this is "hidden away" in Appendix A3. The reason being a thin model domain of only approx 10 m (as Fig 3 and 4 depict) would most likely incur boundary effects from the base boundary condition.

To avoid misinterpretation of our vertical model domain extent, we now specify our lower model boundary in the model-setup section in the main text: "...We used a high vertical resolution for grid cells in the upper 4 meter of the ground and coarse resolution towards our lower model boundary at 1000 m depth, which is subject to an assumed geothermal heat flow of 0.05 W m-2 (Lachenbruch et al., 1982)."

Section 3/Figures 4,5: It would also be nice to see/visualize the boundary conditions which are used to drive the model, if possible, for example the forcing/driving conditions at the surface. This can help understand and interpret the results shown in these Figures.

We now show in a new Supplement Figure (see Fig.2, reply to RC1) the time evolution of the forcing surface air temperature. The forcing shows a gradual increase in surface temperatures which does not explain the thaw dynamics illustrated in Figure 5. When interpreting thaw trajectories in section 3.2, we now refer to the forcing evolution and
focus our discussion on the issue of talik formation (see comments made in response to reviewer 1).

Section 4. The model and approach used in this paper is very nice and clearly relevant, but the purpose of this section is not so clear. It would be far more informative and convincing to actually demonstrate the strategy for the different applications suggested here. For example, can a demonstration of a model linking to LSMs beyond the 100-m scale modeled here be made? Also, the discussion on suggested analysis for risk assessment from line 350 onwards would have more substance if a demonstration could be made.

We have completely revised section 4 to discuss the linkage of PTMs with LSMs and GTMs and now discuss more concretely how a combined modelling strategy could be achieved (see new section 4).

Fig 4: Consider making a note that the vertical extent (elevation) of the model domain is not depicted here, could refer to Appendix A3. Also in Fig 3, the extent of the model in the vertical direction is depicted only as around 10 m, but according to A3 is 1000 m.

We now state in the legends of Fig.3 and Fig.4 that the lower model boundary is at 1000m to avoid misinterpretation of the extent of our vertical model domain.

Fig 6: Please improve clarity, font sizes and legends are relatively small.

We will improve readability of the figure.

Table 2: Hydraulic properties are not described; what is the hydraulic conductivity or permeability, and porosity, of the different materials used? Typically, there is great variability and hence uncertainty in hydraulic conductivity and this greatly impacts water Flow.

We use a constant hydraulic conductivity of 1e-5 m/sec for all soil types. We now added this information in the parameter table. We do not resolve differences in hydraulic conductivities between the natural tundra ground and the subgrade material. As our
considered model site does not provide conditions for large lateral subsurface water flow, we expect comparatively small lateral water exchange rates a realistic description. Porosities for the individual soil and embankment layers are given in table 1.

Table 2: Do the thermal conductivities refer to dry porous media? How is effective thermal conductivity, accounting for phase state of water-ice in the porous media, calculated?

Effective thermal conductivity of each layer is calculated based on the volumetric fractions of the ground constituents water, ice, air, mineral, gravel, and organic (Cosenza et al.,2003; Westermann et al., 2013). The temperature dependence of the effective thermal conductivity is taken into account by calculating temperature-dependent water and ice contents. We now added this information in the table legend of table 2.

Table 3: Is there a 'conservative' setting for the HighRes model setup? Please clarify.

We now added the information in table 3 that for the HighRes model setup no conservative setting was run.

Fig A2: What do the model ranges, min to max, with median, refer to here? What is changed between the "Cryogrid model runs" to get the ranges of results? Please Clarify.

The indicated min-max range illustrates the spread of estimates, each inferred from 11 individual simulations years, covering the period 2010 to 2020. The years are subject to differing climatological input. We now have re-formulated the figure legend to clarify this issue: ...."Blue dots and blue lines illustrate simulated year-to-year variability and indicate the median and min-max range estimated from 11 snow seasons simulated by CryoGrid3 for the period 2010 to 2020,..."

TCD

---

## Author Response (AR1)

We thank the reviewers for their constructive comments which are very helpful for improving our manuscript. Based on the comments made by both reviewers, we have especially reworked our perspective section. We also extended our model simulations to explore long-term model behaviour beyond the year 2100 and investigated further model diagnostics to explore in more detail the model physics discussed, especially with respect to talik formation (see below). In the following we reply in detail to all the comments made:

**Reviewer 1**

...However, two general concerns I have are that: 1) I would like the authors to elaborate more on the outline they propose in section 4 for how to expand simulations like those reported here to a pan-Arctic risk assessment, and 2) I would appreciate a bit more discussion of the permafrost physics in the simulations from their Dalton Highway case study. In addition, I have several more specific comments, listed below.

**General comments:**

1) I appreciate that the authors sketch out how to incorporate PTMs into pan-Arctic risk assessments in section 4, but currently the proposed workflow is somewhat difficult to understand. I know that this section is speculative, but I'd like a bit more detail about the connection between LSMs and PTMs. The authors state that results from an initial set of PTMs "could ... be used by LSMs to define a tiling-based model-setup for describing infrastructure in a reduced but manageable manner." Dtalikoes this mean that a pan-Arctic LSM simulation would be executed, incorporating a sub-grid tiling scheme that accounts for the location of infrastructure? And if so, how would this be more informative than simply using meteorological forcing data derived from ESMs in the initial set of PTMs?

We have intensively reworked section 4 to substantiate our suggested strategy for connecting PTMs with LSMs and GTMs (see new text at the end of this document) where we now stress more clearly the difference in applicability and informative values between LSM and GTM analyses.

We now sketch out three main lines of connecting PTMs and LSMs/ESMs

1) PTMs can support LSM development for defining suitable tiling-based descriptions of infrastructure such that LSMs can be used for calculating large-scale estimates of potential risks regarding infrastructure failure. LSMs would not account for the exact location of infrastructure, but rather for the general presence of infrastructure within a LSM grid cell. Infrastructure in each grid cell is then subject to the climatic conditions taken from the corresponding atmospheric fields of each LSM grid cell (rather than to site-specific local climate).

2) PTMs could be used as an offline add-on in LSMs for capturing infrastructure.

3) PTMs could be run in stand-alone mode driven by climate conditions from an ESM (like done in this study). This allows the accounting of more site-specific aspects (e.g. local soil stratigraphy, bias-corrected climate forcing, etc.)

For details see new section 4 below.

2) I am also curious at what stage of the analysis it is most appropriate to assess uncertainty related to the site-specific manners in which infrastructure may accelerate permafrost thaw. The authors acknowledge that "a specific gravel road can deviate strongly from our assumed setting here." As they state, one source of variability is road geometry, and another source of variability is that gravel roads can cause thaw through a variety of mechanisms, such as low-albedo dust deposition on the tundra or impedance to surface drainage. Another important factor to consider is that, unlike the case study presented here, the effects of these process are often asymmetrical, occurring preferentially on one side of the roadway (e.g., Raynolds et al., 2014; Abolt et al., 2017). It seems like uncertainty in each of these factors would have a big impact on the risk assessment. Is it best to account for it within PTMs, LSMs, or GTMs?

We agree that these factors are all critical for judging infrastructure failure at a given location. For such cases, the capturing of these processes likely requires the use of complex fully coupled thermal-hydrological 3D modelling. If the focus is more on a regional or pan-arctic risk assessment, site-specific factors are of less importance but it will be crucial to describe controlling factors (such as ponding next to a road) which determine overall infrastructure failure on a regional scale. The potential risk will not depend on where exactly infrastructure will fail, but on the occurrence of infrastructure failure under climatic conditions typical for a given grid cell. Such analyses could be performed with PTM-LSM setups.

In the revised section 4 (see end of this document) we also sketch out in more detail how PTMs can support GTM analyses for determining infrastructure failure under climate change.

3) The authors state in the abstract that rates of permafrost thaw adjacent to the highway follow "a two-phase behavior", and in section 3.2, they state that thaw rates sharply increase "once a critical warming of the ground has been reached." This is an important result and I think it would be worthwhile for the authors to elaborate more on when this threshold occurs. For example, does the abrupt increase in thaw coincide with events such as the formation of an open talik, or the onset of surface water pondingâ×A

'Tor does it lag several years behind? Also, does the system stabilize by the end of the simulations?

This information would be informative to the risk assessment, and I also think it would be interesting in the context of other numerical simulations of thermokarst.

We consider these aspects interesting diagnostics for the model result interpretation and have analyzed talik formation dynamics and performed a climate stabilization scenario to investigate long-term thaw dynamics.

Our analyses show that the timing of increase in thawing rates can occur well before full talik formation. Yet the choice of a less strong criterion for soil refreeze allows us to broadly track the two phase behaviour. We achieve this by determining the timing of the formation of a partially unfrozen soil layer, where refreeze does not exceed 25% of pore water (see the timing indicated

by dashed vertical lines in the updated Figure 5 below). In the manuscript we now discuss this aspect as follows:

"... Thawing rates can be split into two periods: a period of slow and gradual increase in maximum thaw depths, followed by a sharp increase in thawing rates once a critical warming of the ground has been reached. The two-phase thawing characteristic cannot be explained through changes in external climate forcing (see Figure A3), but by internal dynamics following the formation of year-round partially unfrozen layers in the ground. We can roughly track the timing of the increase in thawing rates by the first occurrence of a layer with a maximum refreeze of 25% of the pore water (dashed vertical lines, Figure 5). This diagnostic can be seen as a precursor for later full talik formation (i.e. year-round unfrozen conditions).

Figure 5: Temporal evolution of maximum thaw depth (MTD) for all structural units under RCP8.5 warming (SU1: road centre, SU2: embankment shoulder, SU3: toe, SU4: adjacent tundra). The left panel illustrates the conservative case without ponding at the toe, the right panel shows the vulnerable case with ponding and a southern facing road shoulder. Runs show results from the *MedRes* setting with 5 tiles (see table 3) and show annual means (thin lines) and 5 year moving means (thick lines). Dashed vertical lines indicate the timing of the first occurence of a year-round partially unfrozen layer (maximum refreeze up to 25% of pore water). The horizontal black dotted lines indicate the tundra surface and embankment base.

We now also discuss in section 3.2 in more detail the physical reasons for the difference in pace of thaw between the different structural units:

"The outer road edge is destabilized around the year 2060 (2075 without ponding). Talik formation under the road center only occurs after year 2080, but triggers a very pronounced increase in thawing rates (Fig. 5). The large increase in late-century thawing rates under the road center are a consequence of previous continuous warming of subgrade temperatures through lateral heat flux in depth. This heat flux leads to an almost isothermal temperature depth profile under the road close to 0°C (see Fig.4, lowest panels), making the ground strongly vulnerable to further warming."

For analysing long-term consequences of climate stabilization, we have extended our modelsetup and run our model under the assumption that climate will stay stationary at the end of 21st century RCP8.5 climate conditions. In Figure A3 we discuss the evolution of deep soil temperature (at 5m and at 10m depth) together with surface air warming. We now show that tiles affected by infrastructure show continuously deepening thaw depths beyond 2100, while the tundra tile shows stabilization. In the manuscript we discuss this aspect accordingly:

"We further tested how a stabilization of climate at the end of the 21st century would affect the long-term behaviour of ground thaw dynamics. All tiles which are affected by the presence of the gravel road show continuously increasing thaw depths throughout the 22nd century as deep soil layers reach above-zero temperatures. In contrast, the active layer below the tundra stabilizes at depth (Figure A3). This stabilization is realized despite slightly positive mean annual air temperatures after year 2075 (Figure A3, light blue curves) through combined effects from a pronounced reduction in snow insulation (as a consequence of strongly reduced snow heights and shortened snow seasons) and soil surface drying during summer (resulting in a strong increase in summer insulation of the soil surface). This finding shows the key impact of the protective peat layer for subground temperatures and permafrost state, but we underline that our simulated preservation of tundra permafrost also depends on the chosen model setting (e.g. with respect to external climate forcing and to internal model parameterizations)."